
# Iodine chemistry in the chemistry-climate model SOCOL-AERv2-iodine

Arseniy Karagodin-Doyennel[1,2,*], Eugene Rozanov[1,2,3,*], Timofei Sukhodolov[1,2,3,*], Tatiana Egorova[2,*], Alfonso Saiz-Lopez[4], Carlos A. Cuevas[4], Rafael P. Fernandez[4,5], Tomás Sherwen[6,7], Rainer Volkamer[1,8,9,10], Theodore K. Koenig[8,9], Tanguy Giroud[1], and Thomas Peter[1]

[1]The Institute for Atmospheric and Climate Science (IAC) ETH, Zurich, Switzerland
[2]The Physikalisch-Meteorologisches Observatorium Davos/World Radiation Center (PMOD/WRC), Davos, Switzerland
[3]Department of Physics of Earth, Faculty of Physics, Saint Petersburg State University, Saint Petersburg, Russia
[4]Department of Atmospheric Chemistry and Climate, IQFR-CSIC, Spain
[5]Institute for Interdisciplinary Science, National Research Council (ICB-CONICET), FCEN-UNCuyo, Mendoza, Argentina
[6]National Centre for Atmospheric Science, University of York, York, YO10 5DD, UK
[7]Wolfson Atmospheric Chemistry Laboratories, University of York, York, YO10 5DD, UK
[8]Department of Chemistry, University of Colorado Boulder, Boulder, CO 80309
[9]Cooperative Institute for Research in Environmental Sciences, Boulder, CO 80309
[10]Laboratory of Radiochemistry and Environmental Chemistry, Paul Scherrer Institute, 5232 Villigen, Switzerland
[*]These authors contributed equally to this work.

**Correspondence:** Arseniy Karagodin-Doyennel (darseni@student.ethz.ch)

**Abstract.** This paper introduces a new version of the chemistry-climate model SOCOL-AERv2, supplemented by an iodine chemistry module. We conducted three twenty-year-long ensemble experiments to assess the validity of modeled iodine and to quantify the effects of iodine on ozone. The obtained iodine distributions with SOCOL-AERv2-iodine show good agreement with the CAM-chem model simulations and AMAX-DOAS observations. For the present-day atmosphere, the model suggests

the strongest influence of iodine in the lower stratosphere with an ozone loss of up to 30 ppbv at low latitudes and up to 100 ppbv at high latitudes. Globally averaged, the model suggests iodine-induced chemistry to result in an ozone column reduction of 3-4%, maximizing at high latitudes. In the troposphere, iodine chemistry lowers tropospheric ozone concentrations by 5-10% depending on the geographical location. We found that in the lower troposphere, the share of ozone loss induced by iodine originating from inorganic sources is 75% and only 25% by iodine from organic sources, and contributions become similar

at about 50 hPa. We also determined the sensitivity of ozone to iodine applying a 2-fold increase of iodine emissions, which reduces the ozone column globally by an additional 1.5-2.5%. These results constrain the importance of atmospheric iodine chemistry for present and future conditions, even though uncertainties remain high due to the paucity of observational data of iodine species.





# 1 Introduction

The emissions of halogenated compounds containing chlorine and bromine have long been identified by the scientific community as a potential threat to the ozone layer owing to their involvement in catalytic ozone destruction cycles (Solomon, 1999). Recent studies demonstrate the success of the Montreal Protocol and its Amendments in phasing out the emissions of ozone-depleting substances containing chlorine or bromine, indicating also first signs of the ozone layer recovery (Newman et al., 2009; Egorova et al., 2013; WMO, 2018). At the same time, iodine-induced ozone depletion has not been studied ex-

tensively because of the exceedingly low atmospheric iodine concentrations (Saiz-Lopez et al., 2012). Nevertheless, iodine chemistry has been suspected to affect stratospheric ozone (Saiz-Lopez et al., 2015) and slow down the recovery of the ozone layer (Koenig et al., 2020). The ozone depletion efficiency of iodine-containing species on a per-molecule basis is hundreds of times stronger than that of chlorine (Solomon et al., 1994; Gutmann et al., 2018; Koenig et al., 2020) as iodine reservoir species in the stratosphere are much less stable in the presence of sunlight. Nevertheless, iodine-induced ozone depletion is

not regulated under the Montreal Protocol as there is almost no direct anthropogenic production of iodine-containing species (Fuge and Johnson, 2015) and because the mixing ratio of total inorganic iodine in the atmosphere is typically extremely low ($\sim$ 1 parts per trillion by volume (pptv)) and, in contrast to other halogens, assumed to be too low to contribute significantly to the ozone depletion.

The investigation of iodine source gases (ISGs) started in the late 1970s. The main organic source of atmospheric iodine is

found to be iodomethane ($CH_3I$) as suggested by Lovelock and Maggs (1973) and Chameides and Davis (1980). Most organic iodocarbons are produced by photochemical processes that involve phytoplankton and micro/macroalgae, while some terrestrial iodocarbons are emitted by wetland plants (Manley et al., 2007), biomass burning (Akagi et al., 2011), rice paddies (Lee-Taylor and Redeker, 2005), and volcanoes (Bureau et al., 2000; Aiuppa et al., 2005). In fact, the main source region of iodine is the ocean that produces almost all organic and inorganic iodine that enters the atmosphere (Saiz-Lopez et al. (2012) and references

therein). The sea surface temperature (SST) is an important driver of biological activity leading to the release of organic ISGs into the atmosphere. Therefore, the ISGs are produced mostly in the tropics. The 5-6 day lifetime of $CH_3I$ is long enough to allow a small fraction of it to reach the upper troposphere by tropical upwelling with an estimated amount $\sim$ 0.1 pptv in the tropical tropopause layer (WMO, 2018). The other organic ISGs, such as $CH_2I_2$, $CH_2ICl$, $CH_2IBr$, and $C_2H_5I$, are considered to be minor contributors to iodine abundance in the upper troposphere and stratosphere since their lifetimes after volatilization

are just a few hours on average. Garland and Curtis (1981) proposed that besides organic ISGs a production of inorganic ISGs such as hypoiodous acid (HOI) and molecular iodine ($I_2$) through the reaction of near-surface ozone with oceanic iodide ($I^-$) has to be considered. This may be particularly important in the marine boundary layer, where reactions such as $O_3 + 2I^- + 2H^+$ $\rightarrow I_2 + H_2O + O_2$ may release iodine from the sea surface water, possibly responsible for up to 75% of total iodine emissions (Carpenter et al., 2013; Prados-Roman et al., 2015b).These findings were considered in numerical modeling of $HOI/I_2$ fluxes

using a numerical parametrization of sea-surface iodide (MacDonald et al., 2014). Due to growing anthropogenic air pollution and the concomitant increase in surface ozone levels iodine emissions have risen continuously during the past decades. Cuevas et al. (2018) and Legrand et al. (2018) argued that the atmospheric iodine loading must have increased by at least a factor of 3





since the 1950s, because of increasing anthropogenic $NO_x$ emissions resulting in an increase in near-surface ozone. This is in addition to a simultaneous increase in SST due to global warming with correspondingly enhanced metabolic rates of oceanic

biota. While the future surface ozone evolution has a large spread in model projections over the 21$^{st}$ century (Archibald et al., 2020) resulting in a large uncertainty in future iodine emissions, the continuous increase in surface temperatures is predicted to raise tropospheric iodine levels throughout the 21$^{st}$ century based on Representative Concentration Pathway (RCP) scenarios (Iglesias-Suarez et al., 2020). Klobas et al. (2021) showed that in future scenarios the effectiveness of iodine for ozone depletion is poorly sensitive to changes in the state of the stratosphere indicating an increase of relative importance of iodine over

other halogens.

Solomon et al. (1994) presented one of the first studies demonstrating the potential role of iodine chemistry in stratospheric ozone loss using a two-dimensional model of atmospheric chemistry and dynamics. The results implied a high impact of iodine chemistry on stratospheric ozone even in the case of only $\sim 1$ pptv of total iodine concentration in the stratosphere. These model results were later put in perspective by Pundt et al. (1998), who presented results of 9 balloon flights over Scandinavia

and France measuring concentrations of reactive $IO_x$ ($IO_x = IO + I$) of less than 0.1 pptv at the upper troposphere, suggesting only a small influence of iodine chemistry on stratospheric ozone loss. In contrast, Wittrock et al. (2000) found concentrations of IO + OIO between 0.65 and 0.8 pptv over Spitsbergen Island. However, their results referred to point measurements valid at the specific time and location of the balloon measurements, which cannot be considered globally representative. Recent aircraft campaigns using the University of Colorado Airborne Multi-AXis Differential Optical Absorption Spectroscopy (CU AMAX-

DOAS) instrument (Baidar et al., 2013) over the West Pacific during CONTRAST ((Convective Transport of Active Species in the Tropics) in January–February 2014 (Pan et al., 2017)) and over the East Pacific during TORERO ((Tropical Ocean Troposphere Exchange of Reactive Halogen Species and Oxygenated VOC) in January–February 2012 (Volkamer et al., 2015))[1] measured a second maximum in IO mixing ratio of more than 0.1 pptv in the lower TTL that had not been observed before. Yet, globally representative quantitative measurements of iodine species are still not available. Quasi-global IO has been inferred

from the SCIAMACHY satellite measurements (Schönhardt et al., 2008), though with a high level of uncertainty.

Vogt et al. (1996, 1999) proposed a detailed iodine, gas and aqueous phase, chemistry reaction scheme using a box model including iodine species recycling via the uptake of HOI onto sea-salt particles resulting in the production of ICl and IBr (i.e., via $HOI + Cl^- + H^+ \rightarrow ICl + H_2O$, $HOI + Br^- + H^+ \rightarrow IBr + H_2O$). Note that these recycling reactions constitute a net source of bromine and chlorine to the atmosphere, but represent only a change in partitioning for the iodine species. Dix

et al. (2013) hypothesized that the measured increase of IO mixing ratio in the Pacific free troposphere might hint at some unexplored heterogeneous recycling processes from aerosols back to the gas phase in the upper troposphere. (Ordóñez et al., 2012) were the first to use a global chemistry-climate model, namely CAM-chem, to incorporate a comprehensive bromine and iodine chemistry scheme. The global distribution of organic iodocarbons from CAM-chem showed generally good agreement with observations in the marine boundary layer, while Prados-Roman et al. (2015a,b) highlighted the requirement to include

a global oceanic ISGs of approximately 1.9 Tg(I)/yr to reproduce the ubiquitous presence of IO measurements. Later, an updated version of the GEOS-Chem model (Sherwen et al., 2016a) allowed to estimate the total emission of iodine-containing

---

[1]The GV AMAX-DOAS Data can be found here: https://data.eol.ucar.edu/





substances to be 3.83 Tg (I)/year, when using the sea-surface iodide field from Chance et al. (2014). Sherwen et al. (2016c) used the sea-surface values from MacDonald et al. (2014), which possibly underestimates the sea-surface iodide emissions in comparison to observations (Sherwen et al., 2019), suggesting 2.7 Tg (I)/year of total emissions (2.2 Tg (I)/year of inorganic

[HOI+$I_2$] and 0.5 Tg (I)/year of organic emissions). The heterogeneous recycling mechanism involving iodine species on ice was first assumed to be analogous to bromine recycling on ice crystals in the upper troposphere (Aschmann and Sinnhuber, 2013) and implemented in CAM-chem model (Saiz-Lopez et al., 2015). The recycling of iodine reservoirs (HOI and $IONO_2$) on ice crystals was implemented to reproduce the IO observations from Volkamer et al. (2015). Based on a good correspondence of the aircraft measurements and CAM-chem simulations, it was concluded that the increase of IO in the upper troposphere

may be caused by the increase in iodine lifetime due to the heterogeneous recycling on ice. The total gas-phase iodine in the tropical stratosphere ($20^o$N-$20^o$S) can reach 0.25-0.75 pptv range depending on whether the heterogeneous recycling mechanism on ice is involved or not. Also, CAM-chem results indicate that stratospheric iodine could be responsible for up to 30 % of halogen-mediated ozone depletion in the lower tropical stratosphere.

Thus, the current total inorganic iodine injected into the stratosphere is thought to be up to 0.8 pptv (WMO, 2018), i.e. four

times larger than that from previously published assessment ($\sim$ 0.15 pptv) (WMO, 2014). At the same time, the results of Hossaini et al. (2015) obtained using the 3-D chemistry transport model TOMCAT suggest that only $\sim$3 % of stratospheric ozone loss is driven by iodine chemistry and that the long-term impact of iodine on total ozone loss is within 0.5%. However, this modeling approach to ozone loss was performed without taking consideration of the mechanism described in Saiz-Lopez et al. (2015), where the stratospheric iodine level is much higher than in Hossaini et al. (2015). It remains difficult to judge

which results are closer to reality because of the paucity of observations.

This paper introduces the new version of the chemistry-climate model (CCM) SOCOL-AERv2 (Solar Climate Ozone Links coupled to a size-resolving sulfate aerosol module) extended with an iodine chemistry scheme. The main objective of this study is to model the total amount of iodine in the atmosphere and particularly in the stratosphere as well as to further constrain the influence of iodine chemistry on stratospheric ozone depletion. The impact of iodine chemistry on ozone loss is studied here

in two main directions: the effect of iodine chemistry on present day global ozone climatology and the sensitivity of ozone to iodine applying a two-fold increase of iodine emissions. The doubling of iodine emissions that is used here is a scenario that might become reality by the middle of this century if the rise of emissions presented by Legrand et al. (2018) and Cuevas et al. (2018) continued, contrary to the much more moderate increase projected by Iglesias-Suarez et al. (2020).

Section 2 introduces the CCM SOCOL-AERv2 (2.1), improvements which have been made to get the new model version

subsequently referred to as CCM SOCOL-AERv2-iodine (2.2), and contains the description of numerical experiments conducted with the new model version (2.3). Section 3 presents the results of the simulation beginning with the comparison of the simulated iodine against available observations: first, different aspects of iodine effect on present day stratospheric ozone climatology are considered (3.1), and second, the sensitivity of ozone to iodine is presented (3.2). The discussion and summary of the present study are provided in Section 4.





## 2 Model description and conducted experiments

### 2.1 The SOCOL-AERv2 chemistry-climate model

The chemistry-climate model SOCOL-AERv2 is the CCM SOCOLv3 (Stenke et al., 2012; Revell et al., 2015) coupled to a size-resolving sulfate aerosol module (AER) (Weisenstein et al., 1997) along with other important modifications for chemistry and deposition. The AER module of SOCOL was established by Sheng et al. (2015) (CCM SOCOL-AERv1). The CCM SOCOL-AERv1 was substantially updated by Feinberg et al. (2019) (CCM SOCOL-AERv2) with interactive deposition scheme, expanding tropospheric chemistry scheme, and with improved sulfate mass and particle number conservation (less susceptible to numerical diffusion). The SOCOL-AERv2 consists of a dynamical core that is the middle atmosphere version of the spectral transform general circulation model MA-ECHAM5.4 (the Middle Atmosphere version of the European Centre/Hamburg Model version 5.4) (Manzini et al., 2006), which has been interactively coupled to the MEZON atmospheric chemistry-transport module (Model for the Evaluation of OZONe Trends) (Egorova et al., 2003). The coupling takes account of radiative forcing caused by ozone, $H_2O$, $N_2O$, $CH_4$, and CFCs. The MA-ECHAM developed at the Max Planck Institute for Meteorology (Hamburg, Germany) is based on primitive prognostic equations for meteorological parameters such as logarithm of surface pressure, temperature, humidity, vorticity, etc. The advection in MA-ECHAM is regulated by a flux-transform semi-Lagrangian scheme based on mass conservation and shape retention (Lin and Rood, 1996). The standard SOCOL-AERv2 utilizes the Gaussian transform horizontal grid with T42 triangular truncation (64 latitudes $\times$ 128 longitudes) splitting the model space into grid cells of $\sim$ 2.5 $\times$ 2.5 degrees each. The vertical direction model grid consists of 39 levels in the hybrid sigma-pressure coordinate system covering the altitudes ranging from the ground surface and 0.01 hPa or about 80 km. The model time step is 15 min for dynamical and physical processes whereas it is 2 hours for atmospheric full radiation and chemical calculations. The CCM SOCOL-AERv2 uses the prescribed monthly fields of the sea surface temperature (SST) and ice coverage acquired from the Hardley Center dataset (Rayner et al., 2003). MEZON shares the horizontal and vertical spatial resolutions with MA-ECHAM5 and includes 95 chemical species, 215 gas-phase, 16 heterogeneous, and 75 photolysis reactions. For more details on CCM SOCOL-AERv2, see Feinberg et al. (2019).

### 2.2 SOCOL-AERv2 updates to SOCOL-AERv2-iodine

Here, we introduce the SOCOL-AERv2-iodine that is the SOCOL-AERv2 extended with the iodine chemistry module. This module includes 61 gas-phase and 4 heterogeneous chemical reactions involving iodine, boundary conditions for prescribed iodocarbon emissions, interactive inorganic iodine emissions, wet/dry depositions of iodine species augmented with the deposition on sea-salt and sulfate aerosol particles as well as effective uptake (removal)/reactive uptake (recycling) on tropospheric cloud ice. The following section describes all components of the iodine chemistry module in SOCOL-AERv2-iodine. The prescribed organic and interactive inorganic iodine source gases (ISGs) are presented in Table 1.

Table 1: Iodine source gases (ISGs) incorporated in CCM SOCOL-AERv2-iodine

————————





| ISG | Type | Resolution | Reference |
|---|---|---|---|
| $I_2$ | interactive | model time step | Carpenter et al. (2013) |
| HOI | interactive | model time step | Carpenter et al. (2013) |
| $CH_3I$ | prescribed | monthly | Ordóñez et al. (2012) |
| $CH_2I_2$ | prescribed | monthly | Ordóñez et al. (2012) |
| $CH_2ICl$ | prescribed | monthly | Ordóñez et al. (2012) |
| $CH_2IBr$ | prescribed | monthly | Ordóñez et al. (2012) |

Following Ordóñez et al. (2012), the organic iodocarbons have been obtained from the inventory of Bell et al. (2002) for $CH_3I$ and from the 1-D model estimates of Jones et al. (2010) for $CH_2I_2$, $CH_2ICl$, and $CH_2IBr$. Organic ISGs have been parameterized in Ordóñez et al. (2012) by a biogenic chlorophyll-$\alpha$ (chl-$\alpha$) dependent source in the tropical oceans (McClain et al., 2004). In our scheme, organic emissions are prescribed on the monthly basis. To simulate the long-term period, we simply recur organic emissions fluxes at the beginning of every model year, so any interannual variability is not included.

Iodocarbon source fluxes were directly extracted from the GEOS-Chem model (v10-01) at a resolution of a $2 \times 2.5$ degrees including updates to both iodine and bromine chemistry from Sherwen et al. (2016c) and Schmidt et al. (2016), and interpolated on the SOCOL-AERv2-iodine horizontal grid (T42). The inorganic $HOI/I_2$ fluxes are interactively calculated in the model using the numerical parametrization of Carpenter et al. (2013) and global sea surface iodide concentration calculated following MacDonald et al. (2014). To derive $HOI/I_2$ fluxes this parametrization utilizes the model fields of near-surface ozone (at closest

to the surface model level), surface wind speed, and SST.

The parametrization for the emission fluxes of $F_{HOI}$ and $F_{I_2}$ (in nmol m$^{-2}$ d$^{-1}$) as function of the surface ozone mixing ratio $\chi_{O_{3,surf}}$ (in ppbv) are given by

$$F_{HOI} = \chi_{O_{3,surf}} \times (4.15 \times 10^5 \times (M_{I_{aq}^-}^{1/2}/W_s) - (20.6/W_s) - 23600 \times M_{I_{aq}^-}^{1/2}), \qquad (1)$$

$$F_{I_2} = \chi_{O_{3,surf}} \times M_{I_{aq}^-}^{1.3} \times (1.74 \times 10^9 - 6.54 \times 10^8 \times \ln W_s), \qquad (2)$$

where $W_s$ is the surface wind speed (in m s$^{-1}$) and $M_{I_{aq}^-}$ is the sea surface iodide concentration expressed as molarity (mol dm$^{-3}$)

$$M_{I_{aq}^-} = 1.46 \times 10^6 \exp(-9134/T_{ss}) \qquad (3)$$

with sea surface temperature $T_{ss}$ in K.





It is important to note that this parameterization is intended for calculating $I_2$ and HOI fluxes at wind speeds below 14 m $s^{-1}$ (higher wind speeds lead to negative values of fluxes) since it is based on the approximation of measurements and there are no measurements when a storm occurs. Also, it should be mentioned that turbulent mixing of the interfacial layer with bulk seawater reduces the proportion of $I_2$ and HOI evading into the atmosphere therefore fluxes decrease with the wind speed (MacDonald et al., 2014). The obtained inorganic ISGs fluxes in the SOCOL-AERv2-iodine exceed GEOS-Chem and CAM-Chem

emissions by about 50% because SOCOL-AERv2-iodine overestimates near-surface $O_3$ by about 50% compared to other models (Revell et al., 2018). To ameliorate this bias, the surface ozone used within the parameterization was scaled by 0.5. This scaling factor was found to be optimal after comparing the tropospheric ozone from GEOS-chem and SOCOL-AERv2-iodine. This decreased the emission by 50% giving HOI/$I_2$ fluxes comparable: $\sim$ 2 Tg (I)/year in SOCOL-AERv2-iodine and 2.2 Tg (I)/year in GEOS-chem (Sherwen et al., 2016c). Since there is a large uncertainty in inorganic iodine emissions (Chance et al.,

2014; Sherwen et al., 2016c), such a difference in emissions between models is admissible. Iodocarbon source fluxes in both SOCOL-AERv2-iodine and GEOS-chem are identical and correspond to 0.5 Tg (I)/year.

  Chemical compounds involved in the iodine chemistry scheme in SOCOL-AERv2-iodine are presented in Table 2.

Table 2: The iodine species considered in SOCOL-AERv2-iodine

| Species | Molar mass (g mol$^{-1}$) | Henry's law constant (H) (mol m$^{-3}$ atm$^{-1}$) | Temperature dependence of H (K) | Reference |
|---|---|---|---|---|
| I | 126.90 | $7.9 \times 10^1$ | 0 | Burkholder et al. (2015) |
| $I_2$ | 253.81 | $2.8 \times 10^3$ | 3900 | Burkholder et al. (2015) |
| IO | 142.90 | — | — | — |
| OIO | 158.90 | — | — | — |
| $INO_2$ | 172.91 | $3.0 \times 10^2$ | 0 | Ordóñez et al. (2012) |
| INO | 156.91 | — | — | — |
| $IONO_2$ | 188.91 | $\infty$ | 0 | Burkholder et al. (2015) |
| HOI | 143.91 | $4.1 \times 10^5$ | 0 | Badia et al. (2019) |
| HI | 127.91 | $7.3 \times 10^{16}$ | 3190 | Badia et al. (2019) |
| IBr | 206.81 | $2.43 \times 10^4$ | 0 | Wagman et al. (1989) |
| ICl | 162.36 | $1.1 \times 10^5$ | 0 | Wagman et al. (1989) |
| $I_2O_2$ | 285.81 | $\infty$ | 0 | Badia et al. (2019) |
| $I_2O_3$ | 301.81 | $\infty$ | 0 | Badia et al. (2019) |
| $I_2O_4$ | 317.81 | $\infty$ | 0 | Badia et al. (2019) |
| $CH_3I$ | 141.94 | $2.0 \times 10^2$ | 3600 | Burkholder et al. (2015) |
| $CH_2I_2$ | 267.84 | $7.3 \times 10^3$ | 0 | Burkholder et al. (2015) |
| $CH_2I$ | 140.93 | — | — | — |
| $CH_2IBr$ | 220.83 | $2.0 \times 10^3$ | 0 | Hilal et al. (2008) |





| CH$_2$ICl | 176.38 | $2.0 \times 10^3$ | 0 | Hilal et al. (2008) |

A dash means that Henry's law constant is unavailable;

For IONO$_2$ and I$_x$O$_y$ Henry's law constants are assumed to be infinity and represented by a very large number.

Overall, 19 iodine species are included in the current version of the iodine chemistry scheme of SOCOL-AERv2-iodine: 6 iodine source gases (ISGs), 13 product gases (PGs) including two species (HOI and I$_2$), which are both being emitted and chemically produced. Photolysis rates and reaction cross-sections for iodine species were taken from Burkholder et al. (2015). An exception is made only for the photolysis rates of higher iodine oxides species, such as I$_2$O$_2$, I$_2$O$_3$, and I$_2$O$_4$, where we followed the recommendations of Davis et al. (1996) and Vogt et al. (1999), who suggested using photolysis rates of higher

iodine oxides nine times higher than those of Cl$_2$O$_2$. This is a more simplified approach than used in CAM-chem, where cross-sections for I$_2$O$_2$ and I$_2$O$_3$ are adopted from Gómez-Martín et al. (2005) and for I$_2$O$_4$, the used spectrum is measured at the University of Leeds (Saiz-Lopez et al., 2014). In GEOS-chem, the cross-sections for higher iodine oxides are equal to those of IONO$_2$ (Sherwen et al., 2016a) based on the assumption made by Bloss et al. (2010). In our scheme, we checked this possibility too and found that there is no noticeable difference between using IONO$_2$ photolysis rates or 9*Cl$_2$O$_2$ photolysis

rates as substitutes for unknown photolysis rates of higher iodine oxides. The Henry's law constant of IONO$_2$ is taken in SOCOL-AERv2-iodine to be infinity by analogous to BrONO$_2$ and ClONO$_2$ ($1 \times 10^{30}$ mol m$^{-3}$ atm$^{-1}$). The Henry's law constant of I$_x$O$_y$ is infinity and represented by $2.65 \times 10^{18}$ mol m$^{-3}$ atm$^{-1}$ (Badia et al., 2019). The Henry's law constant of INO$_2$ is equal to that of BrNO$_2$ presented in Ordóñez et al. (2012).

The full iodine reaction scheme, composed of 61 gas-phase chemical reactions including 17 photolysis reactions and 4

heterogeneous reactions proceeding on tropospheric cloud ice is presented in Table 3.

Table 3: The list of chemical reactions with iodine included to SOCOL-AERv2-iodine

Part 1: Gas-phase chemical reactions.

| Chemical reaction | A-factor [cm$^{-3}$ molecule$^{-1}$ s$^{-1}$] | E$_a$/R (K) | Reference |
|---|---|---|---|
| I$_2$ + O$_3$ → IO + I + O$_2$ | $3.8 \times 10^{-18}$ | — | Ordóñez et al. (2012) |
| I$_2$ + O$_3$ → OIO + IO | $3.8 \times 10^{-18}$ | — | Ordóñez et al. (2012) |
| I + O$_3$ → IO + O$_2$ | $2.3 \times 10^{-11}$ | 870 | Burkholder et al. (2015) |
| I + HO$_2$ → HI + O$_2$ | $1.5 \times 10^{-11}$ | 1090 | Ordóñez et al. (2012) |
| IO + NO → I + NO$_2$ | $9.1 \times 10^{-12}$ | -240 | Burkholder et al. (2015) |
| IO + HO$_2$ → HOI + O$_2$ | $1.3 \times 10^{-11}$ | -570 | Burkholder et al. (2015) |
| IO + IO → OIO + I | $2.13 \times 10^{-11}$ | -180 | Saiz-Lopez et al. (2014) |
| IO + IO → I$_2$O$_2$ | $3.24 \times 10^{-11}$ | -180 | Badia et al. (2019) |



| | | | |
|---|---|---|---|
| $I_2 + OH \rightarrow HOI + I$ | $1.8 \times 10^{-10}$ | — | Burkholder et al. (2015) |
| $I_2 + NO_3 \rightarrow I + IONO_2$ | $1.5 \times 10^{-12}$ | — | Saiz-Lopez et al. (2014) |
| $I + NO_3 \rightarrow IO + NO_2$ | $1.0 \times 10^{-10}$ | — | Badia et al. (2019) |
| $OH + HI \rightarrow I + H_2O$ | $1.6 \times 10^{-11}$ | 440 | Badia et al. (2019) |
| $HOI + OH \rightarrow IO + H_2O$ | $2.0 \times 10^{-13}$ | — | Saiz-Lopez et al. (2014) |
| $INO + INO \rightarrow I_2 + NO + NO$ | $8.4 \times 10^{-11}$ | 2620 | Burkholder et al. (2015) |
| $INO_2 + INO_2 \rightarrow I_2 + NO_2 + NO_2$ | $2.9 \times 10^{-11}$ | 2600 | Burkholder et al. (2015) |
| $IO + BrO \rightarrow Br + I + O_2$ | $3.0 \times 10^{-12}$ | -510 | Badia et al. (2019) |
| $IO + BrO \rightarrow Br + OIO$ | $1.2 \times 10^{-11}$ | -510 | Badia et al. (2019) |
| $I + BrO \rightarrow IO + Br$ | $1.44 \times 10^{-11}$ | — | Saiz-Lopez et al. (2014) |
| $IO + ClO \rightarrow I + Cl + O_2$ | $1.175 \times 10^{-12}$ | -280 | Saiz-Lopez et al. (2014) |
| $IO + O \rightarrow I + O_2$ | $1.4 \times 10^{-10}$ | — | Burkholder et al. (2015) |
| $O + I_2 \rightarrow IO + I$ | $1.4 \times 10^{-10}$ | 0 | Burkholder et al. (2015) |
| $OH + CH_3I \rightarrow CH_2I + H_2O$ | $2.9 \times 10^{-12}$ | 1100 | Burkholder et al. (2015) |
| $Cl + CH_3I \rightarrow CH_2I + HCl$ | $2.9 \times 10^{-11}$ | 1000 | Burkholder et al. (2015) |
| $IO + DMS \rightarrow SO_2{}^* + I$ | $3.2 \times 10^{-13}$ | 925 | Saiz-Lopez et al. (2015) |
| $CH_2I + O_2 \rightarrow CH_2O + IO$ | $4.0 \times 10^{-13}$ | — | Gravestock et al. (2010) |
| $IO + NO_3 \rightarrow OIO + NO_2$ | $9.0 \times 10^{-12}$ | — | Saiz-Lopez et al. (2014) |
| $OIO + NO \rightarrow IO + NO_2$ | $1.1 \times 10^{-12}$ | -542 | Saiz-Lopez et al. (2014) |
| $IO + OIO \rightarrow I_2O_3$ | $1.5 \times 10^{-10}$ | — | Badia et al. (2019) |
| $OIO + OIO \rightarrow I_2O_4$ | $1.5 \times 10^{-10}$ | — | Badia et al. (2019) |
| $IO + Br \rightarrow I + BrO$ | $2.49 \times 10^{-11}$ | — | Saiz-Lopez et al. (2014) |
| $I + IONO_2 \rightarrow I_2 + NO_3$ | $9.1 \times 10^{-11}$ | -146 | Saiz-Lopez et al. (2014) |
| $IO + CH_3O_2 \rightarrow CH_2O + I + HO_2$ | $2.0 \times 10^{-12}$ | — | Saiz-Lopez et al. (2014) |
| $IO + ClO \rightarrow ICl + O_2$ | $9.4 \times 10^{-13}$ | -280 | Saiz-Lopez et al. (2014) |
| $IO + O_3 \rightarrow OIO + O_2$ | $3.6 \times 10^{-16}$ | — | Saiz-Lopez et al. (2014) |
| $IO + OH \rightarrow HO_2 + I$ | $1.0 \times 10^{-10}$ | — | Saiz-Lopez et al. (2014) |
| $HI + NO_3 \rightarrow I + HNO_3$ | $1.3 \times 10^{-12}$ | 1830 | Saiz-Lopez et al. (2014) |
| $I_2O_2 + M \rightarrow OIO + I + M$ | $2.5 \times 10^{14}$ | 9770 | Ordóñez et al. (2012) |
| $I_2O_2 + M \rightarrow IO + IO + M$ | $1.0 \times 10^{12}$ | 9770 | Ordóñez et al. (2012) |
| $I_2O_4 + M \rightarrow OIO + OIO + M$ | $3.8 \times 10^{-2}$ | — | Badia et al. (2019) |
| $I + NO_2 + M \rightarrow INO_2 + M$ | $k_0 = 3.0 \times 10^{-31}$ | 1.0 | Burkholder et al. (2015) |
| | $k_\infty = 6.6 \times 10^{-11}$ | 0 | |
| $INO_2 + M \rightarrow I + NO_2 + M$ | $9.94 \times 10^{17}$ | 11859 | Badia et al. (2019) |





| IO + NO$_2$ + M → IONO$_2$ + M | k$_0$=7.5 × 10$^{-31}$ | 3.5 | Burkholder et al. (2015) |
|---|---|---|---|
| | k$_\infty$=7.6 × 10$^{-12}$ | 1.5 | |
| IONO$_2$ + M → IO + NO$_2$ + M | k$_0$=5.0 × 10$^{-28}$ | 14120 | Burkholder et al. (2015) |
| | k$_\infty$=1.9 × 10$^{-7}$ | 2.5 | |
| I + NO + M → INO + M | k$_0$=1.8 × 10$^{-32}$ | 1 | Burkholder et al. (2015) |
| | k$_\infty$=1.7 × 10$^{-11}$ | 0 | |

Part 2: Photochemical reactions

| Chemical reaction | Method | Reference |
|---|---|---|
| CH$_3$I + hν → CH$_3$ + I | Look-up table | Burkholder et al. (2015) |
| CH$_2$I$_2$ + hν → CH$_2$I + I | Look-up table | Burkholder et al. (2015) |
| I$_2$ + hν → I + I | Look-up table | Burkholder et al. (2015) |
| IO + hν → I + O | Look-up table | Burkholder et al. (2015) |
| OIO + hν → I + O$_2$ | Look-up table | Burkholder et al. (2015) |
| INO + hν → I + NO | Look-up table | Burkholder et al. (2015) |
| INO$_2$ + hν → I + NO$_2$ | Look-up table | Burkholder et al. (2015) |
| IONO$_2$ + hν → I + NO$_3$ | Look-up table | Burkholder et al. (2015) |
| HOI + hν → I + OH | Look-up table | Burkholder et al. (2015) |
| HI + hν → I + H | Look-up table | Burkholder et al. (2015) |
| I$_2$O$_2$ + hν → I + OIO | Look-up table | Davis et al. (1996) |
| I$_2$O$_3$ + hν → IO + OIO | Look-up table | Davis et al. (1996) |
| I$_2$O$_4$ + hν → OIO + OIO | Look-up table | Davis et al. (1996) |
| IBr + hν → I + Br | Look-up table | Burkholder et al. (2015) |
| ICl + hν → I + Cl | Look-up table | Burkholder et al. (2015) |
| CH$_2$ICl + hν → I + Cl | Look-up table | Burkholder et al. (2015) |
| CH$_2$IBr + hν → I + Br | Look-up table | Burkholder et al. (2015) |

Part 3: Heterogeneous chemical reactions on tropospheric cloud ice.

| Chemical reaction | Reactive uptake coefficient [unitless] | Reference |
|---|---|---|





| | | |
|---|---|---|
| $HOI + HI \rightarrow I_2 + H_2O$ | $\gamma = 0.12$ | Saiz-Lopez et al. (2015) |
| $HOI + HCl \rightarrow ICl + H_2O$ | $\gamma = 0.12$ | Saiz-Lopez et al. (2015) |
| $HOI + HBr \rightarrow IBr + H_2O$ | $\gamma = 0.12$ | Saiz-Lopez et al. (2015) |
| $IONO_2 + (H_2O) \rightarrow HOI + HNO_3$ | $\gamma = 0.1$ | Saiz-Lopez et al. (2015) |

A-factor: the pre-exponential factor;

$E_a$: the activation energy;

R: the universal gas constant;

* $SO_2$ instead of DMSO is used (DMSO is absent in SOCOL-AERv2-iodine);

$k_0$: Low-pressure limit($cm^6 molecule^{-2} s^{-1}$);

$k_\infty$:High-pressure limit ($cm^3 molecule^{-1} s^{-1}$)

We briefly review SOCOL's chlorine and bromine reactions because of their interaction with iodine chemistry. Apart from reactions involving iodine, there are about 100 gas-phase and 11 heterogeneous reactions on sulfates and polar stratospheric clouds (PSCs) for chlorine species and about 50 gas-phase and 4 heterogeneous reactions on sulfates and polar stratospheric

clouds (PSCs) for bromine species. The total gas-phase chlorine ($Cl_{tot}$) and bromine ($Br_{tot}$) in the current version of the model are:

$Cl_{tot}$ = ClO + Cl + 2 × $Cl_2$ + $ClNO_3$ + HOCl + HCl + 2 × $Cl_2O_2$ + BrCl + CFC-11 + 2 × CFC-12 + 3 × CFC-113 + 2 × CFC-114 + CFC-115 + 4 × $CCl_4$ + 3 × $CH_3CCl_3$ + HCFC-22 + 2 × HCFC-141B + HCFC-142B + H-1211 + $CH_3Cl$ + 2 × HCFC-21 + 2 × HCFC-123 + HCFC-31 + ICl + $CH_2ICl$ and

$Br_{tot}$ = Br + BrO + 2 × $Br_2$ + BrCl + $BrNO_3$ + HBr + 2 × $CH_2Br_2$ + $CH_3Br$ + $CBrF_3$ + 3 × $CHBr_3$ + HOBr + H-1301 + H-1211 + 2 × H-2402 + IBr + $CH_2IBr$ correspondingly.

Similar to the iodine schemes in CAM-chem (Ordóñez et al., 2012) and GEOS-Chem (Sherwen et al., 2016a), we implemented the free molecular transfer approximation of McFiggans et al. (2000). This allows introducing the iodine scavenging and deposition on sea-salt and sulfate aerosols as well as effective ice-uptake (removal)/reactive ice-uptake (recycling) on a

surface of tropospheric cloud ice crystals (Fernandez et al., 2014; Saiz-Lopez et al., 2014, 2015). The transfer coefficient ($s^{-1}$) is calculated as follows:

$$k = \frac{1}{4}\gamma\langle c \rangle A, \qquad (4)$$

where $\gamma$ is the effective/reactive uptake coefficient (see Table 3), A is the surface area density (in $cm^2\ cm^{-3}$) of the particles on which the deposition occurs, $\langle c \rangle = (8RT/\pi\ M)^{1/2}$ is the mean thermal molecular speed (in $cm\ s^{-1}$) of molecules with molar

mass M (in $Kg\ mol^{-1}$) at absolute temperature T (in K), and R = 8.3145 $J\ mol^{-1}\ K^{-1}$.





In SOCOL-AERv2-iodine, sea-salt aerosols are prescribed by monthly means from observational data and aqueous sulfuric acid aerosols are calculated interactively; from both, surface area densities (SADs) are available. However, there is no readily available SAD for cloud ice in SOCOL-AERv2-iodine. Therefore, we calculate the effective radius $R_{eff}$ (in mkm) of ice crystals following Heymsfield et al. (2014):

$R_{eff} = \alpha \exp(\beta T_c),$          (5)

where $T_c$ is the temperature (in $C^o$), $\alpha = 154.2$ and $\beta = 0.0152$ for $-56 C^o < T < 0 C^o$; $\alpha = 4.5872 \times 10^4$ and $\beta = 0.117$ for $-71 C^o < T < -56 C^o$; $\alpha = 41.65$ and $\beta = 0.0184$ for $-85 C^o < T < -71 C^o$. Following Holmes et al. (2019), the SAD for ice particles (in $cm^2$ $cm^{-3}$) is calculated from ($R_{eff}$) as follows:

   $SAD_{ice} = 6.75 \times IWC/(\rho R_{eff}),$          (6)

where $\rho$ is the density of ice ($9.167 \times 10^{-4}$ kg $cm^{-3}$) and IWC is the ice water content, i.e. the mass of ice per volume of air in the cloud (in kg $cm^{-3}$).

Since the timescale of the physical process of removing/recycling of iodine is shorter than the model time step (2 hrs), using an explicit integration scheme may result in excessive removal/recycling of iodine species leading to errors (such as negative concentrations). To avoid this, we decided to implement a simple implicit scheme:

$C_1 = C_0/(1 + K \times \Delta t)$          (7)

Where: $C_0$ - the initial concentration; $K$ – transfer coefficient; $C_1$ – the final concentration; dt - model time step for chemistry. This scheme avoids producing negative $C_1$.

As effective uptake coefficients ($\gamma$), we applied $\gamma_{IONO_2} = 0.01$, $\gamma_{INO_2} = 0.02$, $\gamma_{HOI} = 0.06$, $\gamma_{I_2O_2} = \gamma_{I_2O_3} = \gamma_{I_2O_4} = 0.01$ on sea-salt aerosols (Ordóñez et al., 2012).

Since $\gamma$'s for sulfate aerosols are currently unknown, for sulfate particles, $\gamma$'s for sea-salt aerosols were divided by 100 (as the amount of iodine to be removed on tropospheric sulfate aerosols is assumed to be 100 less than for sea-salt).

For effective ice-uptake of iodine, $\gamma$'s are taken to be the same as in CAM-chem (Saiz-Lopez et al., 2014): $\gamma_{HOI} = 0.0003$; $\gamma_{IONO_2} = 0.005$; $\gamma_{HI} = 0.02$. The removal on all presented surfaces is operated only within the troposphere (it is confined by the tropopause level). It must be mentioned that the values of effective uptake coefficients for iodine species for different

types of surfaces are highly uncertain (Saiz-Lopez et al., 2014). It is worth saying that we only consider the deposition on sea-salt and sulfate aerosols and did not implement any heterogeneous recycling of iodine species on sea-salt aerosols like those described in Vogt et al. (1996, 1999), Ordóñez et al. (2012), and Saiz-Lopez et al. (2014). In the heterogeneous recycling mechanism on ice, reactive uptake coefficients for cloud ice are taken as follows: $\gamma_{HOI} = 0.12$; $\gamma_{IONO_2} = 0.1$ (see Saiz-Lopez et al. (2015) supplements). Transfer coefficients for heterogeneous reactions (see Table 3) are also calculated by free molecular



transfer approximation (McFiggans et al., 2000) but using reactive uptake coefficients. The reactive ice-uptake and recycling of HOI and $IONO_2$ is applied in SOCOL-AERv2-iodine after effective ice-uptake and removing of HOI, $IONO_2$, and HI. The sequence of removing/recycling processes is unclear but the chosen sequence shows reasonable results.

## 2.3    Conducted experiments

To evaluate the iodine chemistry scheme of SOCOL-AERv2-iodine as well as to estimate the influence of iodine chemistry

on ozone we designed and carried out three transient numerical experiments. The first one is the control experiment where iodine emissions set to zero. For the second experiment ($1 \times$ iodine), we applied a basic configuration with the present-day iodine emissions. These experiments are used to evaluate the veracity of iodine in SOCOL-AERv2-iodine and to estimate the influence of iodine chemistry on present-day ozone climatology.

To assess whether the potential intensification of iodine emissions in the future will have a tangible effect on the ozone layer,

we designed a sensitivity experiment (to verify the sensitivity of ozone to iodine) in which all iodine emissions are doubled to present-day emissions ($2 \times$ iodine). In essence, it could be considered as a worst-case scenario due to a huge discrepancy between scenarios for the future evolution of iodine precursors like tropospheric ozone (Archibald et al., 2020) and SST (Taylor et al., 2012), despite no dramatic forecast of iodine emission's evolution was made by Iglesias-Suarez et al. (2020). The sensitivity of ozone to increase of the iodine emissions we characterized by comparing experiments with $2 \times$ and $1 \times$ load-

ing of iodine. All experiments were run for the 1990-2009 period including the 10-year spin-up (1990-1999) from the initial conditions that is necessary for iodine to reach the quasi-equilibrium state. The spin-up period was excluded from further analysis. Each experiment consists of 10 ensemble members with a one-month perturbation of initial $CO_2$ concentration to get 10 different atmospheric realizations and to calculate the statistical significance of the iodine effect using the t-Student test. The summary of the experimental set-up can be found in Table 4.


Table 4: The experiments with SOCOL-AERv2-iodine

| Name of experiment | Experiment description | Period of simulation and spin-up |
| --- | --- | --- |
| $0 \times$ iodine | Control run (10 ensemble members) | 1990-2009 (1990-1999 spin-up) |
| $1 \times$ iodine | Present-day emissions (10 ensemble members) | 1990-2009 (1990-1999 spin-up) |
| $2 \times$ iodine | Doubled emissions (10 ensemble members) | 1990-2009 (1990-1999 spin-up) |





## 3 Results of simulation

### 3.1 Evaluation of the iodine from SOCOL-AERv2-iodine against CAM-chem and AMAX-DOAS observations

The total gas-phase inorganic $I_y$ both $2 \times$ iodine and $1 \times$ iodine experiments of SOCOL-AERv2-iodine averaged over tropics
[20°N - 20°S], for the 2000-2009 period and 10 ensemble members is presented in Figure 1.

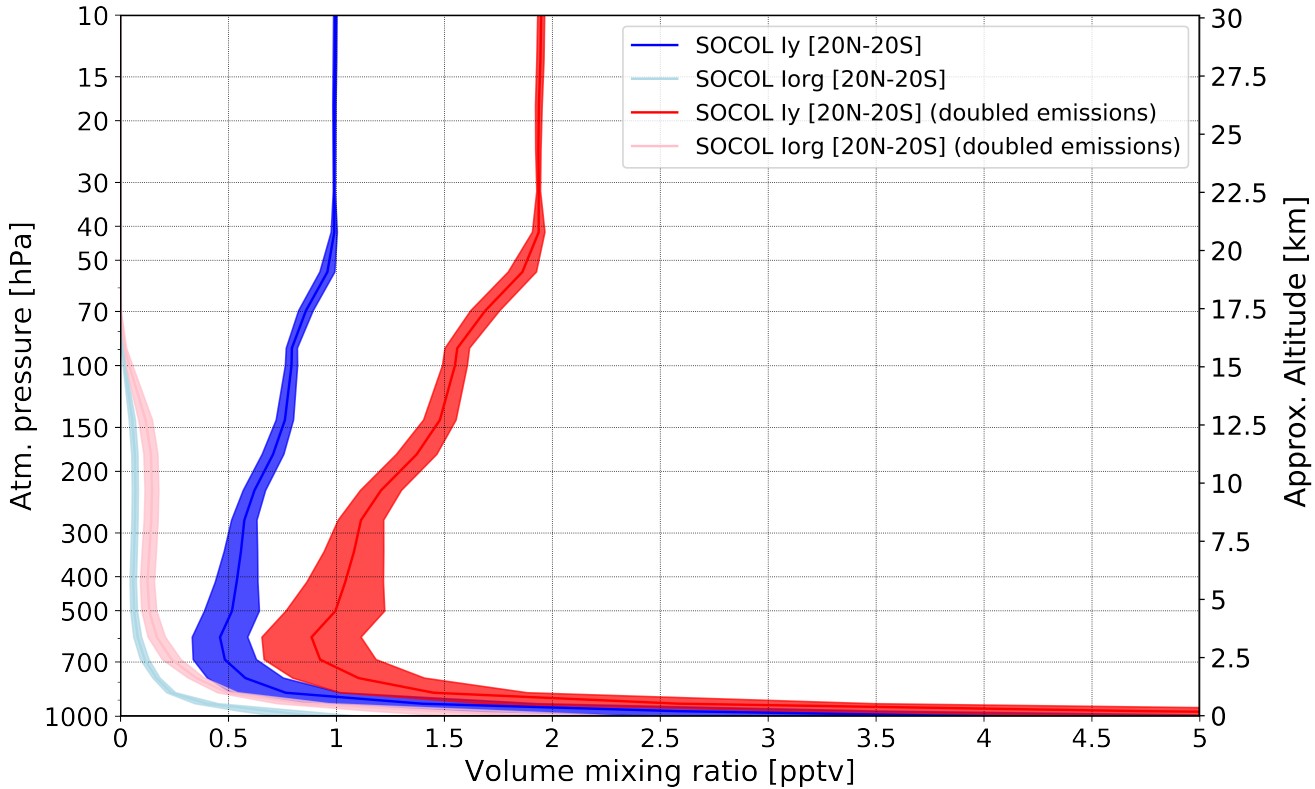

**Figure 1.** Modeled vertical distribution of total organic ($I_{org}$) and inorganic ($I_y$) gas-phase iodine simulated with SOCOL-AERv2-iodine averaged over tropics [20°N - 20°S], for 2000-2009 period and 10 ensemble members. Red curve: $I_y$ from the experiment $2 \times$ iodine. Blue curve: $I_y$ from the experiment $1 \times$ iodine. Light red and blue curves: $I_{org}$ from $2 \times$ iodine and $1 \times$ iodine experiments, correspondingly. Shadings represent a standard deviation of tropical ozone [20°N - 20°S].

The $I_y$ was calculated as a sum of inorganic iodine compounds presented in the SOCOL-AERv2-iodine's iodine scheme: $I_y = IO + I + OIO + HOI + 2 \times I_2 + HI + INO + INO_2 + 2 \times (I_2O_2 + I_2O_3 + I_2O_4) + ICl + IBr + IONO_2 + CH_2I$. $I_{org}$ is calculated as a sum of all organic compounds in SOCOL-AERv2-iodine: $I_{org} = CH_3I + 2 \times CH_2I_2 + CH_2ICl + CH_2IBr$. In the lower troposphere, the $I_y$ is rapidly dropping with altitude until about 600 hPa following the washout that is most effective at
this layer. In the upper troposphere (above 200 hPa), $I_y$ increases after the initiation of the recycling heterogeneous mechanism





on ice that works at these altitudes at tropics because the availability of cloud ice is essential. The recycling on cloud ice mostly defines the amount of inorganic iodine injected into the stratosphere because it competes with washout of reservoir species by converting reservoirs into species with lower washout rates and thus increasing the residence lifetime. The stratospheric $I_y$ shows an increase until about 50 hPa and then stays constant because there is no deposition of iodine above this layer.

The simulated stratospheric $I_y$ by SOCOL-AER2-iodine agrees well with the results of CAM-chem model in the lowermost stratosphere showing about 0.75-0.8 pptv of $I_y$, however it becomes higher in the middle stratosphere than in Saiz-Lopez et al. (2015). It might have resulted from peculiarities of model non-conservative transport scheme and dynamic, for example, the deep tropical convection cells over the area of iodine production that overcomes the deposition velocity enhancing the stratospheric iodine loading. The gradual increase that seen in the stratosphere, could be also attributed to the tropospheric sinks that

define the vertical profile since the lower part of the conservative zone still might be affected by these sinks resulting in some accumulation before the $I_y$ in the conservative zone becomes invariable. A hard upper border in SOCOL-AERv2-iodine that prevents chemical species from going through it also might impact the profile. Nevertheless, the stratospheric $I_y$ abundance calculated with SOCOL-AER2-iodine does not exceed 1 pptv and corresponding well to the estimation given by Solomon et al. (1994), although is slightly larger than the most recent assessment from (WMO, 2018) (0.8 pptv $I_y$).


We compare the iodine monoxide (IO) obtained by SOCOL-AER2-iodine with the one from CAM-chem model and the recent aircraft observations with AMAX-DOAS conducted during the TORERO and CONTRAST campaign (Volkamer et al., 2015; Pan et al., 2017). The results are shown in Figure 2.





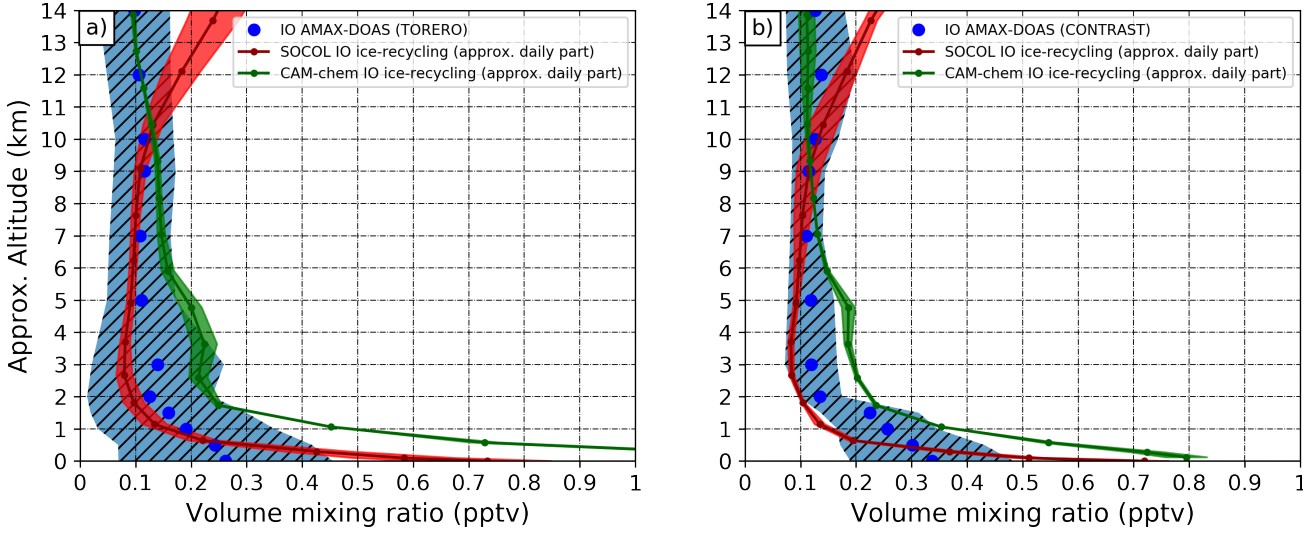

**Figure 2.** January/February averages of modeled and observed IO in the tropical troposphere for (a) the TORERO campaign from Costa Rica (Jan./Feb. 2012, 10°N-40°S, 250°E-285°E), and (b) the CONTRAST campaign from Guam (Jan./Feb. 2014, 40°N-15°S, 115°E-175°E). Red line: IO from SOCOL-AERv2-iodine. Green line: IO from CAM-chem. Blue dots: IO observed by AMAX-DOAS. Shadings: IO standard deviations of all modeled/measured IO during the January-February period.

Modeled IO in Figure 2 has been obtained by doubling monthly averaged IO as during nighttime the IO concentration is negligible. Also, the observations are given as so-called footprints to get profiles for each day, they were around-averaged for certain altitudes designated in Figure 2 as blue points. For example, the first point is a mean of all measurements obtained between 0-0.1km. Both SOCOL-AERv2-iodine and CAM-chem overestimate observations at near-surface levels. The sharp decrease of observed and modeled IO from both models goes similarly until 2 km. However, the modeled IO from CAM-chem over both regions is higher than IO from the SOCOL-AERv2-iodine. IO from the SOCOL-AERv2-iodine is close to the mean of observations for both regions until about 7-9 km. After 10 km, SOCOL-AERv2-iodine overestimates observed IO over both regions whereas IO from CAM-chem became closer and even almost fit them. The sharp increase in SOCOL-AERv2-iodine IO probably following the recycling activity on ice. There may be several reasons for this: the recycling of iodine species on ice in SOCOL-AERv2-iodine starts working at lower latitudes than in CAM-chem, recycling on ice is much more efficient than washout in SOCOL-AERv2-iodine above 10 km, whereas in CAM-Chem they are comparable or tropospheric ozone above 10 km in SOCOL-AERv2-iodine is much larger than in CAM-Chem. In CAM-chem the sharp increase in IO concentration is seen after 15 km (Saiz-Lopez et al., 2015).

Besides, we made a comparison of modeled $I_y$ from SOCOL-AERv2-iodine and CAM-chem models with the $I_y$ derived from IO/$I_y$ ratio modeled with the University of Colorado (CU) chemical box-model constrained by measured temperature, pressure, chemical concentrations, particle size distributions, and photolysis frequencies. The uncertainty in derived $I_y$ is estimated as





30% of the IO/$I_y$ ratio including errors in the calibration of in-situ and remote sensing data and accounting for differences in the spatial scales (Wang et al., 2015; Koenig et al., 2017). IO was taken as an average of IO fields measured with AMAX-DOAS during both TORERO and CONTRAST campaigns (Volkamer et al., 2015; Pan et al., 2017; Koenig et al., 2020). Thus, the inferred $I_y$ is based on the measured IO and modeled IO/$I_y$. The result of this comparison is illustrated in Figure 3.

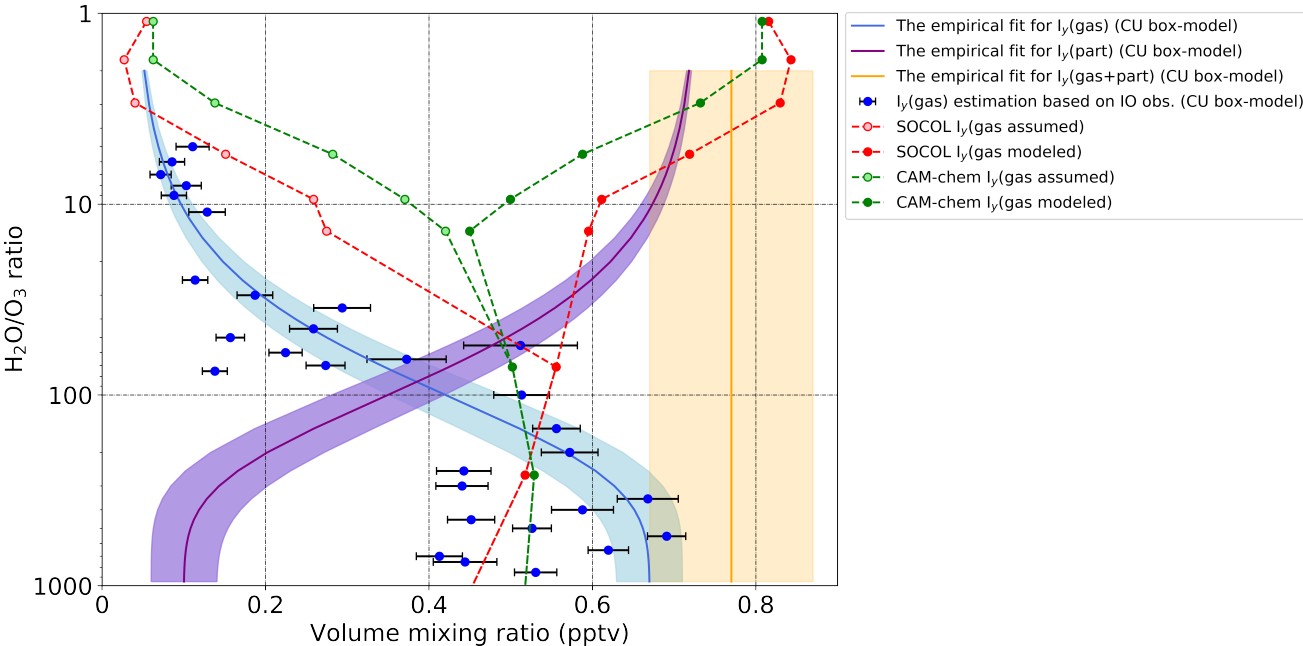

**Figure 3.** Total gas-phase $I_y$ modeled with SOCOL-AERv2-iodine and CAM-chem as well as estimated by CU chemical box-model based on AMAX-DOAS IO observations in the upper troposphere and lowermost stratosphere. All model data and observations are taken only for the region of TORERO [10$^o$N-40$^o$S, 250$^o$E-285$^o$E] and CONTRAST measurements [40$^o$N-15$^o$S, 115$^o$E-175$^o$E]. Red and green solid and dashed lines: gas-phase $I_y$ simulated with SOCOL-AERv2-iodine and CAM-chem averaged for the January-February period. Light red and light green dashed lines: assumed gas-phase $I_y$ simulated with SOCOL-AERv2-iodine and CAM-chem excluding appox. iodine in the particulate phase. Blue dots: gas-phase $I_y$ modeled by the University of Colorado (CU) chemical box-model. Black error bars: the uncertainty in $I_y$ modeled by the CU box-model. Blue, purple, and orange solid lines represent the empirical fit for gas-phase $I_y$ (gas), particulate $I_y$ (part) and total $I_y$ (gas+part), correspondingly. Shadings represent the uncertainty in the empirical fit.

Data were plotted with respect to the $H_2O/O_3$ ratio as it is a good indicator to distinguish the upper troposphere (UT) air enriched in water vapor from the dehydrated air in the lower stratosphere as proposed by Koenig et al. (2020). To make CU box-model data clearer they were thinned out by averaging the data every 50 units of $H_2O/O_3$ ratio between 1000 and 100 of $H_2O/O_3$ ratio; every 5 units between 10 and 1; and every 1 unit between 10 and 1. The blue empirical fit is to a subset of the blue dots, that the orange line is based on $I_y$ (gas) and $I_y$ (part) data in the upper troposphere, and the purple line is derived

under the assumption of conversion. Equations to calculate empirical fit lines for experimental data can be found here (see





equation 2, 4, and 5 in Koenig et al. (2020) supplements).

Below the $H_2O/O_3$ ratio $\sim$70, both models well-capture the $I_y$ estimated from observations. Higher up, the red and green dashed lines represent the gas-phase $I_y$ that is simulated with both models. Because the particulate iodine is not considered in these models, the total inorganic $I_y$ in SOCOL-AERv2-iodine and CAM-chem presented here is only in gas-phase. The light

red and green lines are the assumed gas-phase $I_y$ if the estimated particulate iodine is excluded from the modeled $I_y$. To exclude the unknown particulate iodine, the modeled gas-phase $I_y$ was subtracted from the 0.87 pptv of iodine that is empirical total $I_y$ plus uncertainty of observations (0.1 pptv) (see orange solid line and its uncertainty in Figure 3). Such an approach can show what would be the approximate modeled gas-phase $I_y$ if the iodine in the particulate phase was presented in models. It is seen that after the level of $H_2O/O_3$ ratio $\sim$ 70, the gas-phase $I_y$ estimated from measurements rapidly decreases and reaches the

value of about 0.1 pptv at the approximate altitude of the lowermost stratosphere ($H_2O/O_3$ ratio < 10). After excluding the estimated particulate iodine, the assumed modeled gas-phase $I_y$ becomes very close to the one from CU box-model.

It bears mentioning that there is evidence that a certain part of gas-phase iodine undergoes partitioning to aerosol in the stratosphere. This mechanism is not fully understood due to a lack of measurements. Note that the approach used here is different from the simplified parameterization of IPART in CAM-Chem (Koenig et al., 2020). The assumption that the overestimation

of modeled gas-phase $I_y$ against observations is because of an absence of iodine in the particulate phase is reasonable as it is seen in Figure 3. In this work, we do not consider particulate iodine for the analysis of ozone loss as it is out of the scope of the paper. Here, we assume that the total stratospheric $I_y$ is only in the gas-phase. Nevertheless, the total amount of iodine obtained with SOCOL-AERv2-iodine is in a good agreement with other estimates and observations and can be used in further analysis of its effect on ozone.

**3.2    Iodine chemistry contribution to the global ozone loss**

We estimate the iodine effect on present-day ozone climatology by comparing the experiment with a single loading of iodine (1 $\times$ iodine), and the control experiment neglecting iodine chemistry (0 $\times$ iodine). The contribution of iodine chemistry to present-day ozone climatology estimated by the SOCOL-AERv2-iodine is presented in Figure 4.

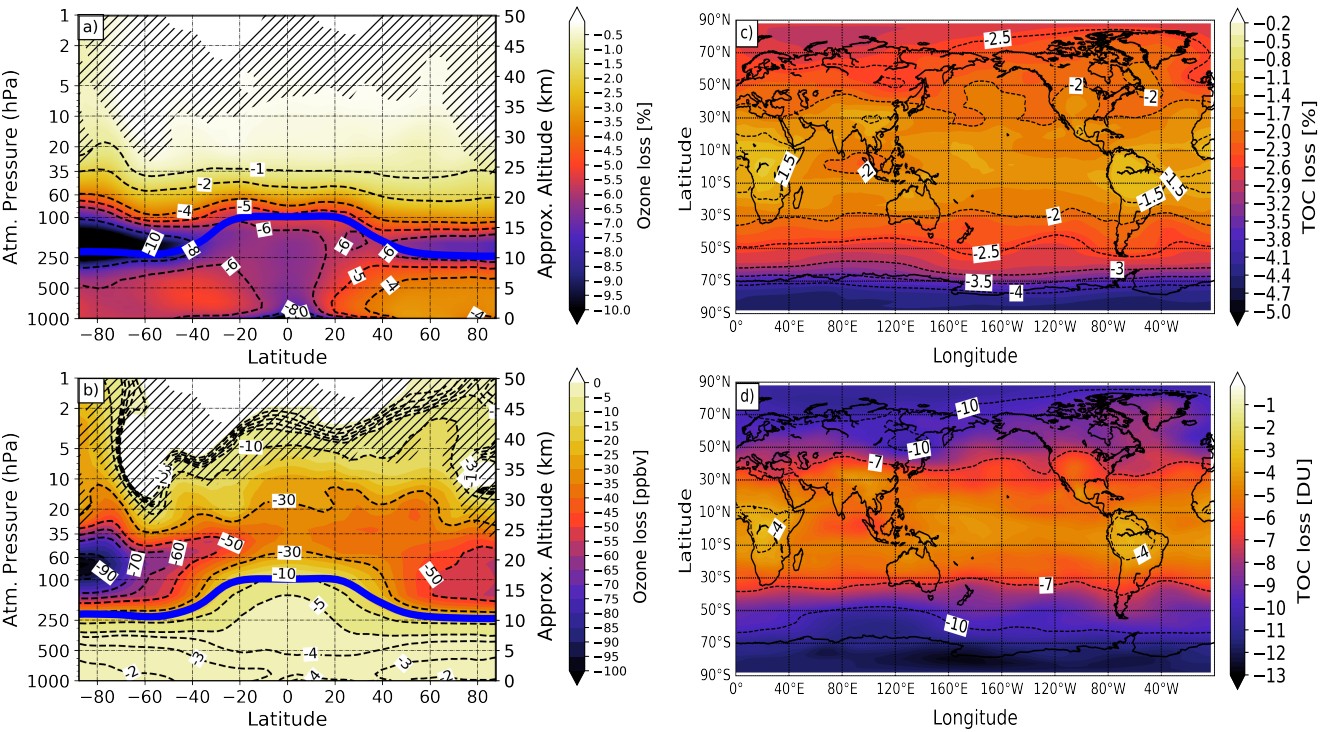

**Figure 4.** Modeled effect of iodine chemistry on annual-mean ozone climatology averaged for 2000-2009 period and 10 ensemble members. (a+c) Ozone changes of the case with present-day iodine emissions (1 × iodine) relative to the control run (0 × iodine), presented as zonal mean ozone mixing ratios and total ozone columns (TOC) in percent. (b+d) Corresponding absolute ozone changes in parts per billion by volume (ppbv) and Dobson units (DU), respectively. Blue solid line in (a) and (b): annual mean tropopause height. Hatching marks regions with ozone losses having a confidence level less than 95%.

The relative and absolute responses of ozone to iodine chemistry activation are shown for both ozone mixing ratio and total ozone column (TOC). The relative iodine effect on ozone is calculated as follows: ((EXP-REF)/REF)*100, where EXP is the ozone from the experiment with 1 × loading of iodine and REF is the control run without iodine being included. The absolute difference is simply defined as EXP-REF. The crisp iodine signal in ozone mixing ratio is observed in the lower stratosphere and intensifies over the polar regions where the effect of halogens is estimated to be higher (Chipperfield et al., 2018). The peak of ozone loss resides in the lower stratosphere over the high southern latitudes, where the ozone loss reaches about 10% or 100 ppbv. The hemispheric asymmetry of the effect over high latitudes might be caused by a difference in the mixing ratio of chlorine and bromine in active form (ClO and BrO) that in turn react with iodine. The effectiveness of iodine also depends upon the coupling with chlorine and bromine chemistry intensifying their ozone depletion cycles in the lower stratosphere (see chemical reactions in Table 3).There is evidence that cross cycles between bromine and chlorine are the dominant contribution to ozone loss in the mid-latitudes and polar region (Fernandez et al., 2017; Alejandro Barrera et al., 2020). The active iodine in





the form of IO is also an effective reaction partner for the BrO+ClO as suggested by Saiz-Lopez et al. (2015). Thompson et al. (2015) assumed that the interaction of IO with the BrO and ClO might be more effective over high latitudes because of their higher concentration in that region (Sioris et al., 2006).

The iodine-mediated ozone loss in the tropical lower stratosphere ranging between 4–5% or 20 ppbv below 20 km. The ozone destruction caused by iodine chemistry is not so pronounced over the tropics because the extremely low temperature in the

cold trap on the tropical tropopause resulted from adiabatic cooling is even lower than the temperature over high latitudes at the same height making catalytic cycles less effective.

It is important to mention that the effects of iodine chemistry are confined mostly to the lower stratosphere and are greatly decreasing from the middle to the upper stratosphere. The "glasses-like" iodine effect in the upper stratosphere similar to the one of $ClO_x$ - cycle (Zubov et al., 2013) is not observed. The $IO_x$-catalytic cycle is ineffective in the upper stratosphere

similarly to $BrO_x$-cycle despite the presence of atomic iodine up there in reasonable concentration. It might be because iodine reservoirs are much more unstable than those of chlorine and, similarly to bromine, the iodine species in lower altitudes will be more likely in active form (IO) than that of chlorine will be in form of ClO (Daniel et al., 1999). So, the probability of the terminal reaction ClO+O is higher than those of IO+O or BrO+O in the upper stratosphere. Hence, the effect of iodine chemistry on upper stratospheric ozone loss is negligibly small.

The tropospheric effect is ∼ 4-5 ppbv and maximizes over tropics where iodine sources are mostly emitted from the ocean thanks to its higher temperature. About 6-8% of tropospheric ozone loss is comparable to what was reported by Sherwen et al. (2016a) but where surface iodine emissions were a bit higher. It is also in agreement with an estimation made by Davis et al. (1996). Tropospheric ozone loss in SOCOL-AERv2-iodine is a bit higher than in CAM-chem where the iodine-induced ozone loss does not exceed 2-3 ppbv (Saiz-Lopez et al., 2014).

The total ozone column (TOC) is affected by iodine mostly over high latitudes (see c) and d) panels of Figure 4). The highest impact of iodine on climatological TOC is over high latitudes of the Southern Hemisphere showing the TOC loss about 4% or 11-12 DU. Over the Northern Hemisphere, the iodine effect on TOC does not exceed 3%.

### 3.3 Ozone response to the increased iodine emissions

To estimate the consequences for ozone from the continuous increase of iodine emissions, we compare ozone from the exper-

iment with doubled emissions to that from the experiment with observed (or present-day) emissions. The results are shown in Figure 3.3.

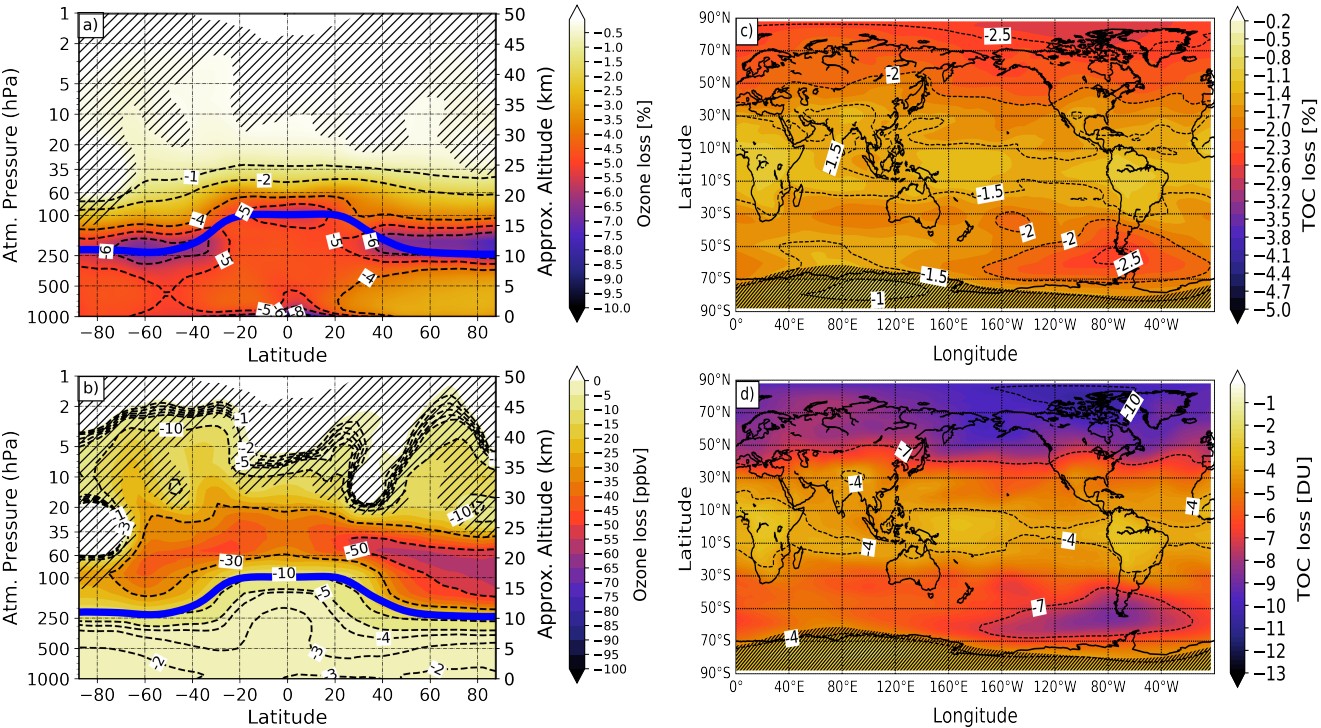

**Figure 5.** Modeled effect of 2-fold iodine chemistry on annual-mean ozone climatology averaged for 2000-2009 period and 10 ensemble members. (a+c) Ozone changes of the case with 2-fold iodine chemistry (2 × iodine) relative to present-day iodine emissions (1 × iodine) in percent. (b+d) Corresponding absolute ozone changes in parts per billion by volume (ppbv) and Dobson units (DU), respectively. Blue solid line in (a) and (b): annual mean tropopause height. Hatching marks regions with ozone losses having a confidence level less than 95%.

In this case, the EXP was taken to be the experiment with 2 × loading of iodine whereas the REF is the one with observed (1 ×) iodine emissions. In the troposphere, a 2-fold increase in emissions leads to the ozone loss of about 6-8% that is similar to what is seen in Figure 4. In the stratosphere, the contribution of additional iodine is different. The Southern hemispheric

maximum is weakened and shifted to the middle latitude showing the ozone loss of up to 7% or 50 ppbv. The iodine contribution to ozone loss in the northern polar lowermost stratosphere also shows a similar pattern and magnitude of the effect to climatological effect in Figure 4 showing about 50 ppb of ozone loss. For the Northern Hemisphere, the effect might be characterized as a linear-kind too and the intensification of ozone loss by a factor of 2 can be expected. The iodine-induced total column ozone loss would enhance by 2-3% following the 2-fold increase of iodine emissions.


Thus, we can expect that a 2-fold increase in iodine injection into the atmosphere would lead to a mostly linear increase of the ozone loss over the troposphere and lower stratosphere. The reason why we do not see the linearly changed effect over the Southern Hemisphere is possibly related to the saturation effect when the ozone was almost destroyed even for smaller iodine



loading. Hence, it can be predicted that the iodine effect on ozone in the lower atmosphere, if the assumed negative iodine

scenario plays out in the future, would simply hinge on a factor of the increasing iodine injection into the atmosphere.

### 3.4    The relative importance of inorganic versus organic iodine sources for iodine-induced ozone loss

We also address the impact of organic vs inorganic iodine sources on total iodine-induced tropical ozone loss. To do this, we repeated the experiment with observed (or present-day) emissions but nullifying either organic or inorganic surface emissions. The results are shown in Figure 6.

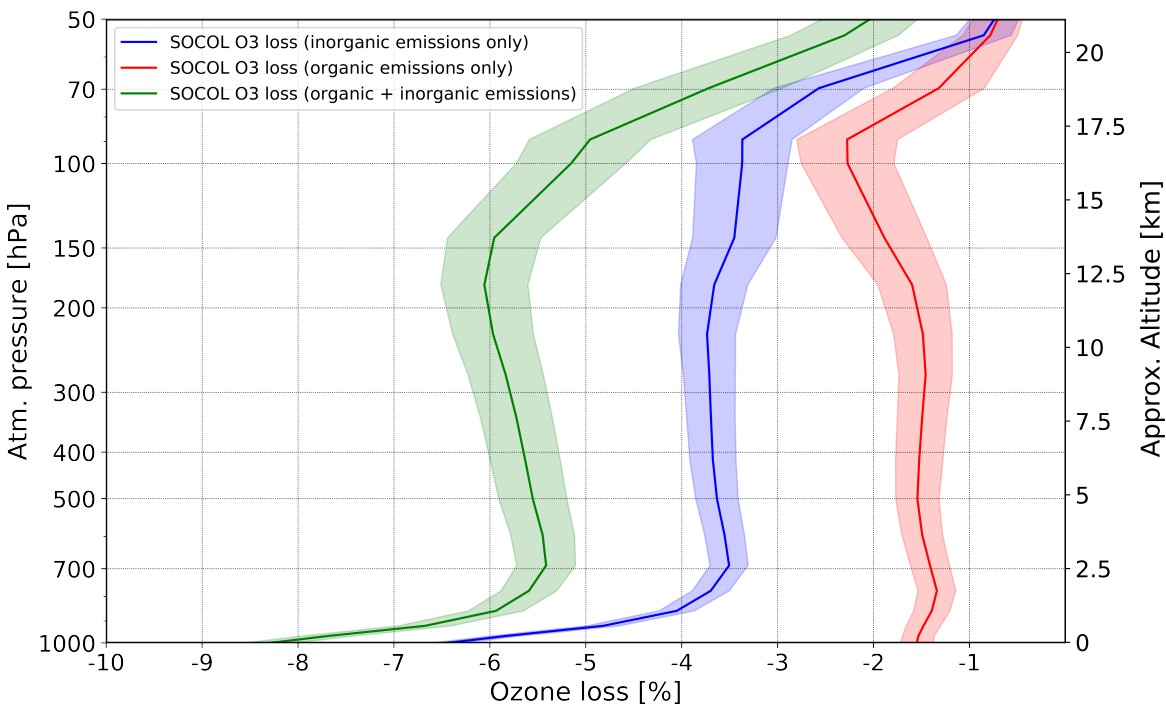

**Figure 6.** Modeled vertical distribution of ozone loss induced by iodine from organic, inorganic, and total surface emissions averaged over tropics [20°N - 20°S], for 2000-2009 period and 10 ensemble members. Red curve: iodine-induced $O_3$ loss, if organic emissions are only considered; Blue curve: iodine-induced $O_3$ loss, if inorganic emissions are only considered; Green curve: iodine-induced $O_3$ loss, if both, organic and inorganic emissions are considered. Shadings represent a standard deviation of ensemble members. The results have a confidence level more or equal to 95%.

In the lower troposphere, the iodine from inorganic emissions is responsible for $\sim 75\%$ of total ozone loss, and the contribution of iodine from organic sources is $\sim 25\%$, as expected (Koenig et al., 2020). In the upper troposphere and lower stratosphere, the ozone loss by iodine from both kinds of sources is closer but still with the higher impact of inorganic sources showing $\sim 60\%$ and $\sim 40\%$, correspondingly. Contributions of iodine both origin to total ozone loss become similar at about 50 hPa.



## 4 Discussion and conclusions

In this study, we describe the new version of the chemistry-climate model SOCOL-AERv2 improved with the addition of the iodine chemistry module. The iodine chemistry scheme in SOCOL-AERv2-iodine was developed based on the up-to-date knowledge about atmospheric iodine. We performed a set of numerical experiments to test the fidelity of the developed iodine chemistry scheme and to estimate iodine contribution to ozone depletion. The model results show about 0.75 pptv of iodine in the lowermost tropical stratosphere in agreement with previous estimations Saiz-Lopez et al. (2015). A gradual increase of $I_y$ up to 1.0 pptv in the stratosphere might be related to the dynamical features, effect of tropospheric sinks or rigid upper border in the model atmosphere. The comparison of modeled and observed IO within the tropical troposphere showed that IO from SOCOL-AERv2-iodine is in a generally good agreement with AMAX-DOAS observations. The additional comparison of total gas-phase inorganic $I_y$ with the values determined by the CU box-model based on AMAX-DOAS observations of IO showed that the model reproduces the $I_y$ in the upper troposphere very well, while in the stratosphere, $I_y$ is much overestimated due to the absence of simulated iodine in particulate phase. If the assumed particulate iodine was excluded from the modeled stratospheric $I_y$, SOCOL-AERv2-iodine corresponds well with $I_y$ from the CU box-model. The simulation of particulate iodine is the subject for future studies as the mechanism of its formation is still not fully understood due to a lack of measurements.

The estimated contribution of iodine chemistry on the lower stratospheric ozone is higher than those discussed in Hossaini et al. (2015) showing up to 10% of the lower stratospheric ozone loss driven by iodine chemistry. It should be noted that Hossaini et al. (2015) reported only 0.15 pptv of $I_y$ injected into the stratosphere, which is more than five times less than in SOCOL-AERv2-iodine and CAM-chem. In the lower troposphere, the share of ozone loss induced by iodine originating only from inorganic sources is estimated to be 75% and 25% if considering only organic sources. Contributions of iodine from organic and inorganic sources to total iodine-induced ozone loss become similar at 50 hPa. We also verified that even if the concentration of iodine is much less than other halogens it could play a noticeable role in the lower stratospheric ozone depletion especially over high latitudes. Nevertheless, negative lower stratospheric changes recently found by Ball et al. (2018) might be driven by iodine chemistry only in the lowermost stratosphere as the iodine effect in the extratropical lower stratosphere is not supposed to produce such a spread signal. The upper stratospheric ozone is not affected by iodine chemistry similar to the impact of bromine chemistry as presumably, the iodine species in the upper stratosphere are hardly being in active form (IO) than that of chlorine will be in form of ClO due to longer lifetime of chlorine precursors (Daniel et al., 1999). Hence, the net effect is that iodine is relatively more important in the lower stratosphere despite the abundance of total gas-phase iodine is expected to be similar throughout the stratosphere.

Also, we would like to address here some shortcomings of the current iodine scheme and further updates that are anticipated to increase the accuracy of iodine simulations in chemistry-climate models. A strong local decrease in free tropospheric ozone can be expected from iodine collected inside iodized aerosol particles from deserts and oceans which can reach the above-cloud troposphere where they release iodine (Volkamer et al., 2021). However, such complex aerosol iodine chemistry





cannot be properly simulated with current chemistry-climate models (Baker, 2004; Sherwen et al., 2016b). The recent studies
of Gómez-Martín et al. (2020), Baccarini et al. (2020), and He et al. (2021) suggested that iodine species can make new aerosol
particles that are big enough to be cloud condensation nuclei (CCNs). Polar ice-melting in turn leads to the increase of the
atmospheric amount of iodine (Cuevas et al., 2018) which may enhance the formation of CCNs. These findings are worth
studying further using global climate models with advanced aerosol chemistry and cloud microphysics. Additionally, higher
iodine oxides species presented in our study by $I_2O_2$, $I_2O_3$, and $I_2O_4$ might decompose to form iodine oxoacids that growth
further to become CCNs (McFiggans et al., 2004; Burkholder et al., 2004; Saunders et al., 2010). However, the formation of
iodine oxoacids is still an open question that needs further studying (He et al., 2021). Here, we did not consider any iodine in
aerosol form, and these species are represented only in the gas-phase. Also, as was mentioned above, in this work we used a
simplified approach for photolysis of higher-order iodine oxides. However, the cross-sections for photolysis of these species
are recently measured and can be used to increase the accuracy of simulation (Lewis et al., 2020). Recent field evidence indi-
cates that the recycling of iodine on sea-salt aerosol, and perhaps other aerosol, may be much faster than currently represented
(Tham et al., 2021). Also, we did not include $CF_3I$ that also could modify the total concentration of iodine in the atmosphere.
However, based on the recent studies, it will not substantially impact the stratospheric ozone loss showing even less impact
than that of $CH_3I$ (Zhang et al., 2020) and will mostly affect the tropical and northern mid-latitudes tropospheric ozone because
of higher concentration of pollutants (Youn et al., 2010). All organic iodine emissions in our scheme are prescribed. However,
they can be interactively calculated in the model utilizing ocean biogenic sources (Ordóñez et al., 2012). It can be embodied
also in the next-generation Earth system model where the ocean biosphere is interactively calculated. As it was mentioned
above, the sea-surface iodide that is the precursor for $HOI/I_2$ fluxes has a huge uncertainty in the models and observations
(Chance et al., 2014; Sherwen et al., 2019). Nonetheless, there is a prediction for increased iodide that will potentially impact
the iodine abundance in the future (Carpenter et al., 2021). Also, there are region-specific parameterizations for sea surface
iodide concentration that can be implemented in the next version of the iodine scheme in SOCOL-AERv2-iodine to increase
the reliability of abiotic iodine emissions (Inamdar et al., 2020). Recent investigations of the chemical basis of the $HOI/I_2$
source can help improve the generalization of empirical source functions (Moreno et al., 2020).

Our sensitivity study showed that the contribution of increased iodine to ozone is almost linear compared to the present-day
iodine. To simulate the reliable future impact of iodine on ozone, the recent estimations on future iodine emissions based on
RCP scenarios can be used (Iglesias-Suarez et al., 2020). Tropospheric ozone content in SOCOL-AERv2-iodine is overesti-
mated compared to other models (Revell et al., 2018) that affects the sea-surface deposition of $O_3$ and its concentration inside
the marine boundary layer, hence the accuracy of simulated iodine emissions. It is planned to fix this problem in future versions
of SOCOL.

One of the most controversial parts of atmospheric iodine studies is the scrutiny of the role of volcanic iodine in stratospheric
chemistry and its effect on ozone. The volcanic iodine impact is worth studying since some of the powerful volcanoes are sup-
posedly capable to directly inject the iodine into the stratosphere and suppose to result in negative and long-lasting implications
for the ozone layer (Bureau et al., 2000; Aiuppa et al., 2005; Balcone-Boissard et al., 2010; Cadoux et al., 2015). However,
there is no solid evidence that volcanoes can inject a sufficient amount of iodine into the atmosphere (Schönhardt et al., 2017)



and it is needed to organize the measurement campaigns to make estimations of emitted volcanic iodine in a more precise way.
The results of this work showed the highest impact of iodine on ozone in the lowermost stratosphere at high latitudes. This finding indicates the necessity of having broad measurements of iodine species in this region.

We also stress that the iodine can presumably be more noteworthy in the future. The intensified Brewer-Dobson circulation might bring more iodine into the stratosphere in the future than today. Also, the further increase of sea surface temperature due to global warming, near-surface ozone, and sea surface iodide concentrations could be the reason for the intensification of
485 iodine emissions in the future making the atmospheric amount of iodine to be vastly higher (Cuevas et al., 2018; Legrand et al., 2018; Cuevas et al., 2018; Koenig et al., 2020; Iglesias-Suarez et al., 2020; Carpenter et al., 2021). The effectiveness of iodine for ozone destruction is found to be stable in future warming scenarios and therefore its relative importance increases relative to the other halogens (Klobas et al., 2021).

All of this inspires further efforts to better characterize the iodine in the atmosphere and its impact on ozone loss.
Alongside, the further improvements of iodine chemistry simulations in chemistry-climate models, it is needed to overcome the scarcity of global measurements of iodine chemistry especially in the upper troposphere and lower stratosphere to increase the accuracy of estimations for iodine impact on ozone loss and to make better predictions of the future ozone evolution.

*Code and data availability.* The SOCOL-AERv2-iodine code is available here: https://doi.org/10.5281/zenodo.4844994 (Karagodin-Doyennel, 2021a) upon request to the corresponding author. The SOCOL-AERv2-iodine simulation data can be accessed here:
https://doi.org/10.5281/zenodo.4820523 (Karagodin-Doyennel, 2021b). CU-AMAX-DOAS CONTRAST IO data are available at: (https://data.eol.ucar.edu/dataset/383.023, last access: 09 June 2021). CU-AMAX-DOAS TORERO IO data are available at: (https://data.eol.ucar.edu/dataset/352.082, last access: 09 June 2021). The CU-box model data are available here: https://doi.org/10.5281/zenodo.4916787 (Volkamer and Koenig, 2021).

*Author contributions.* AK-D conducted all simulations, visualized the results, and drafted the manuscript. ER, TSu, TE, and TG analyzed
the simulations. AS-L, CA.C, RP.F provided the CAM-chem data and assisted the analysis. TSh provided iodine organic fluxes for boundary conditions. RV and TK.K provide CU-box model data. This study was conceptualized and supervised by ER and TP. All authors participated in the model development, discussions about the results, and contributed to writing and editing the manuscript.

*Competing interests.* The authors declare that they have no conflict of interest.

*Acknowledgements.* A.K.-D., E.R., T.S., and T.E. express gratitude to the Swiss National Science Foundation for supporting this research through the №200020-182239 project POLE (Past and Future Ozone Layer Evolution). R.V. acknowledges funding by the US National





Science Foundation (awards AGS-2027252, AGS-1261740, AGS-1104104). R.V. is currently an ETH guest professor, and recipient of a Swiss National Science Foundation fellowship (award 199407). Authors thank Center for Climate Systems Modeling (C2SM) for their support and ETH's High Performance Computing Center (ID SIS) for the possibility to use the Euler Linux cluster to conduct our numerical experiments.



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
