# Peer review of "Iodine chemistry in the chemistry-climate model SOCOL-AERv2-I"

_Geoscientific Model Development, 2021_

## Author Comment (AC1)

Dear reviewer,

We wish to thank you for taking the time to review our paper and provide constructive comments, response to which have greatly helped improve our manuscript.

In the following we present the referee's comments (in bold) and our associated responses to each comment.

**- I would suggest the authors provide more information for their simulation results in Section 3. I think at least a map for Iy or IO should be provided in a similar way as Fig.4. Based on that, a detail explanation of how model mechanisms result in the simulated distribution could be given. Such information can largely help the analysis in later sections.**

Thank you for your comment. We agree that the figure with vertical distributions and global maps of $I_y$ and IO is worth providing in the paper to support some of the assumptions given in the paper. The figure and description are presented below.

[Figure]

Figure 1. - Modeled vertical distribution and global map of total inorganic gas-phase iodine ($I_y$) and iodine monoxide (IO) simulated with SOCOL-AERv2-I from the experiment with present-day iodine emissions (1 × iodine), averaged for 2000-2009 period and 10 ensemble members. (a+b) zonal-mean vertical distributions of $I_y$ and IO. (c+d) global maps of $I_y$ and IO averaged for 100-70 hPa region. Black solid line in (a) and (b): annual mean tropopause height.

Figure 1 presents the vertical distributions of $I_y$ and IO at all latitudes and global maps of $I_y$ and IO averaged for the lower stratosphere (100-70 hPa). $I_y$ and IO have the highest mixing ratio at the polar regions of the stratosphere due to transport carried out by Brewer–Dobson circulation (BDC). The larger stratospheric values of IO at high latitudes are also related to the higher $O_3$ abundance at those locations. At the tropics, there are two pronounced areas with a higher (over Indian ocean) and with a lower (over South America) $I_y$ mixing ratio as depicted in figure 1 c). Their formation might have resulted from different convection patterns (weaker/stronger) over these areas. Also, the area with a higher stratospheric $I_y$ burden is located right over the region with higher surface emissions of $HOI/I_2$ (see figure 2). In the troposphere, the iodine level is decreasing toward the poles and far from

the iodine source regions. The iodine distribution demonstrates the highest mixing ratio of iodine in the lower stratosphere over middle-to-high latitudes with a maximum $I_y$ of more than 1.15 pptv in the polar region of the Northern hemisphere and about 1 pptv in the Southern hemisphere. The IO has two maximums of about 0.4 pptv in the lower-to-middle stratosphere and high latitudes. Also, note that IO decreases at higher levels of the stratosphere exhibiting more than four times less abundance than in the lower stratosphere, which might result from decreasing the efficiency of O3-I reactions.

**- Section 2.2, figures for emission fluxes could be helpful, maybe can be shown in supplements.**

Yes, you are right, such a figure will give some insight into the global distribution of iodine source fluxes. We decided to add the figure depicting both organic [$CH_3I$, $CH_2I_2$, $CH_2ICl$, $CH_2IBr$] and inorganic [$HOI/I_2$] annual-mean fluxes to the main text (presented below).

[Figure]

Figure 2 - Annual-mean emission fluxes of organic [$CH_3I$ (a), $CH_2I_2$ (b), $CH_2ICl$ (c), and $CH_2IBr$ (d)] and inorganic [HOI (e) and $I_2$ (f)] iodine source gases [in nmol m$^{-2}$ d$^{-1}$] used in SOCOL-AERv2-I.

**- I think the introduction is way too long. As a manuscript for model development, it spends nearly two pages to introduce atmospheric iodine. I would suggest shorten the introduction and refer to the references for the details.**

We are not quite sure that the introduction is too long. We think that this kind of paper is worth describing the main up-to-date knowledge and mechanisms related to the behavior of iodine in the atmosphere, including the current state of numerical simulation of iodine to support the importance of this study. However, we agree that some wordings can be changed to decrease the text length. So, we revised the introduction and shortened it by about 15-20% by excluding and reformulating some sentences.

**- Line 170, is the 50% underestimation similar everywhere? Will applying the scale factor make the distribution of the emissions different from other models?**

We reformulated this description to be clearer. Actually, the SOCOL's ground-level ozone is not uniformly 2 times higher than that from the GEOS-chem model. In Revell et al., (2018), the comparison of ozone from SOCOL to another model and observations is provided. It should be noted that a positive bias of ozone is also exhibited in other models e.g. ACCESS, EMAC-L47, and MRI-ESMr1 (see Revell et al., (2018)). Based on results provided in Revell et al., (2018), we chose to apply the 2 times decrease of the ground-level ozone in SOCOL (that used within the $HOI/I_2$ parameterization). We still have higher emissions than those in GEOS-chem, but this difference is laid within their uncertainty. It should be mentioned that this scaling factor does not affect the distribution of $HOI/I_2$ emissions, only the released amount is changed.

Reference:

Revell, L. E., Tummon, F., Stenke, A., Sukhodolov, T., Coulon, A., Rozanov, E., Garny, H., Grewe, V., and Peter, T.: Drivers of the tropospheric ozone budget throughout the 21st century under the medium-high climate scenario RCP 6.0, Atmospheric Chemistry & Physics, 15, 5887–5902, https://doi.org/10.5194/acp-15-5887-2015, 2015.

**- Line 241, any reference or potential evidence for this assumption?**

Since there is no available information about the deposition rate of iodine species on sulfate aerosols and if consider uptake on sulfate aerosols with the same gamma as for sea-salt aerosols, we would obtain modeled $I_y$ values with a rather large bias (X %) against observations. So, after several model tests, we decided to apply sea-salt gammas divided by 100 to simulate the effective uptake and removal of iodine on sulfate aerosols, which brought the iodine from SOCOL-AERv2-I to be closer to available observations.

**- Will the sea salt alkalinity affect the uptake of iodine species?**

According to available studies, there is no evidence on the effect of alkalinity on the uptake rates of iodine species as the understanding of uptake rates of iodine species on sea-salt aerosols is still poor (Saiz-Lopez et al. 2012). However, it is well known that heterogenous halogen reactions, including iodine, are impacted by acidity. This will impact the number of source species released (e.g. Br and Cl) or cycled to more labile forms (e.g. I) This is an important point when considering the changes in halogen burdens and impacts over time and the subject of work in the field, for instance, Zhai et al (2021).

Reference:

Saiz-Lopez, A., Plane, J. M. C., Baker, A. R., Carpenter, L. J., Von Glasow, R., Gómez Martín, J. C., McFiggans, G., and Saunders, R. W.: Atmospheric Chemistry of Iodine, Chem. Rev., 112, 1773–1804, https://doi.org/10.1021/cr200029u, 2012.

Zhai, S., Wang, X., McConnell, J. R., Geng, L., Cole-Dai, J., Sigl, M., et al. (2021). Anthropogenic impacts on tropospheric reactive chlorine since the preindustrial. Geophysical Research Letters, 48, e2021GL093808. https://doi.org/10.1029/2021GL093808

**- Section 2.2, it seems that the model doesn't include reactive uptake of iodine in liquid clouds. Is this because there is no enough information available for the model parameterization? Maybe make it clearer in the text.**

As far as it is currently known, there is no evidence of reactive uptake of iodine species in liquid clouds. The mechanism of heterogeneous reactive uptake of iodine on ice crystals is also still not fully explored and was only proposed as an analogous to that involving chlorine and bromine (Saiz-Lopez et al. 2015). Thus, in our model, we applied the reactive uptake of iodine on ice crystals as was done in Saiz-Lopez et al. (2015).

Reference:

Saiz-Lopez, A., Baidar, S., Cuevas, C. A., Koenig, T. K., Fernandez, R. P., Dix, B., Kinnison, D. E., Lamarque, J. F., Rodriguez-Lloveras, X., Campos, T. L., and Volkamer, R.: Injection of iodine to the stratosphere, grl, 42, 6852–6859, https://doi.org/10.1002/2015GL064796, 2015.

**- Figure 1, maybe make the line thicker and the shadings lighter, or use the logscale. It is very hard to see the values in the lower levels.**

Yes, thank you, we agree that solid lines in this figure are too thin. We made them thicker. The updated figure is added to the paper (presented below).

[Figure]

Figure 3. Modeled vertical distribution of total organic ($I_{org}$) and inorganic ($I_y$) gas-phase iodine simulated with SOCOL-AERv2-I averaged over tropics [20$^o$N - 20$^o$S], for 2000-2009 period and 10 ensemble members. Red curve: $I_y$ from the experiment 2 x iodine. Blue curve: $I_y$ from the experiment 1 x iodine. Light red and blue curves: $I_{org}$ from 2 x iodine and 1 x iodine experiments, correspondingly. Shadings represent a standard deviation of tropical iodine [20$^o$N - 20$^o$S].

**- Line 355-362, this part needs to be explained clearer. The authors referenced a few evidences to show the contribution of cross cycles, but didn't point out whether those evidences were proved in their own model simulation. Maybe provide some numbers to describe the contributions of sole iodine cycles and cross cycles on ozone loss would help.**

Thank you for this comment. We agree that the numbers of what is the role of cross-reactions of iodine with bromine and chlorine in the total iodine-induced effect on ozone are worth mentioning here. To do this, we designed the additional model experiment where all cross-reactions of iodine with Br and Cl are excluded from the overall iodine scheme (reaction rate coefficients are set to zero). A % difference in ozone between experiments with/without cross-reactions included related to the no-iodine experiment is presented in figure 4.

[Figure]

Figure 4 - Modeled effect of iodine chemistry on annual-mean ozone climatology including (left panel) / excluding (right panel) I-Cl and I-Br cross-halogen reactions, averaged for 2000-2009 period. Presented ozone changes of the case with present-day iodine emissions (1 × iodine) relative to the control run (0 × iodine). Black solid line: annual mean tropopause height.

Based on Figure 4, we can conclude that in SOCOL the role of cross-reactions maximizes at the middle and higher latitudes of the lower stratosphere that agrees well with Fernandez et al., (2017) and Barrera et al., (2020). We found that I-Cl and I-Br cross-reactions are responsible for about 60-70% of the total effect of iodine on lower stratospheric ozone in the Northern hemisphere and about 40% in the Southern hemisphere.

Thus, cross-cycles of iodine with Br and Cl themselves are supposed to play an important role in ozone reduction than just chemical reactions of iodine with ozone.

Reference:

Fernandez, R. P., Kinnison, D. E., Lamarque, J.-F., Tilmes, S., and Saiz-Lopez, A.: Impact of biogenic very short-lived bromine on the Antarctic ozone hole during the 21st century, Atmospheric Chemistry & Physics, 17, 1673–1688, https://doi.org/10.5194/acp-17-1673- 2017, 2017.

Barrera A., J., Fernandez, R. P., Iglesias-Suarez, F., Cuevas, C. A., Lamarque, J.-F., and Saiz-Lopez, A.: Seasonal impact of biogenic very short-lived bromocarbons on lowermost stratospheric ozone between 60° N and 60° S during the 21st century, Atmospheric Chemistry & Physics, 20, 8083–8102, https://doi.org/10.5194/acp-20-8083-2020, 2020.

**- Line 372, similar as above, the authors referenced Daniel et al. (1999) for the explanation, but did the model simulation in SOCOL show the same?**

To demonstrate that the suggestion provided in Daniel et al. (1999) is reasonable for our model results we can refer to Figure 1. In Figure 1, we show the zonal-mean vertical distribution of IO between 1000-1 hPa. This figure demonstrates that the IO mixing ratio sufficiently decreases from the lower to the upper stratosphere. At the upper stratosphere, the mixing ratio of IO is more than four times lower than in the lower stratosphere. This resulted from decreasing the efficiency of catalytic cycles involving iodine. We think that it well supports the argument set out in Daniel et al. (1999).

Reference:

Daniel, J. S., Solomon, S., Portmann, R. W., and Garcia, R. R.: Stratospheric ozone destruction: The importance of bromine relative to chlorine, , 104, 23,871–23,880, https://doi.org/10.1029/1999JD900381, 1999.

---

## Author Comment (AC2)

Dear reviewer,

We wish to thank you for taking the time to review our paper and provide inputs and constructive comments, response to which helped us much to improve our manuscript.

In the following we present the referee's comments (in bold) and our associated responses to each comment.

**This paper describes the implementation of a troposphere-stratosphere iodine scheme in the SOCOL CCM. The scheme is based on the well-used CAM Chem scheme. I think that after the suggested revisions the paper will largely serve the purpose of documenting the model iodine scheme and will be suitable for publication in GMD.**

**I thought that the evaluation of the model was somewhat superficial and some of the text was not clear. The whole paper would benefit from a thorough proof-reading (see some examples below).**

**Specific Comments**

**(1) General. Use of the word 'loss'. The manuscript mentions ozone 'loss' throughout, including the abstract (line 5). Really, what is shown is the ozone difference between a model run with and without iodine. The word loss (like 'depletion') to me implies some time trend or change. (There is also photochemical production/loss rates but that is not what is being shown either). If there was no change in the iodine emissions then this difference would always be present (subject to trends in reaction partners), it was just that models were not so accurate without it. So, I suggest reading through the paper and being clear what is being shown by the difference between the model experiments.**

Thank you for your comment and for having read our paper carefully. We accept the point that using the "ozone loss" term without proper context might be unclear while reading the text. So, we edited the paper and reworded it in most of all places to make the text more clear for comprehension by readers.

**(2) Line 6-7. Confusing because the number range quoted is globally averaged so we have no idea of the maximising value at high latitude.**

Yes, you're right, this sentence was poorly formulated. We corrected this sentence as follows:

"For the present-day atmosphere, the model suggests that the iodine-induced chemistry leads to a 3-4% reduction in the ozone column, which is greatest at high latitudes."

**(3) Line 10. Confusing because the sentence appears to be about the lower troposphere but then discusses 50 hPa. Maybe change 'and' to 'but' and explicitly state that 50 hPa is in the stratosphere.**

Thank you. Agreed - this sentence is a bit confusing. We revised this sentence and split it into two separate sentences as follows:

"In the lower troposphere, 75% of the modeled ozone reduction originates from inorganic sources of iodine, 25% from organic sources of iodine. At 50 hPa, the results show that the impacts of iodine from both sources are comparable."

**(4) The importance of iodine (or not) depends not just on how much ozone might be destroyed by iodine but by any time trend in the abundance. I don't think these results 'constrain' anything – they show the sensitivity.**

Thank you. Yes, the word "constrain" might not fit here well. We reformulated this sentence according to your comment as follows:

"Our results demonstrate the sensitivity of atmospheric ozone to iodine chemistry for present and future conditions, but uncertainties remain high due to the paucity of observational data of iodine species."

**(5) Line 135. Hadley Centre (spellings)**

It was fixed.

**(6) Line 149. Use of word 'recur' not clear to me.**

In our model, boundary conditions for all organic iodine sources are directly from the GEOS-chem model. They are one-year long and have the monthly-mean temporal resolution. Under the word 'recur' we meant that these fluxes are repeating each model year of simulation. Maybe just the word "repeat" will be more clear?

**(7) Line 207. 3 x CFC11**

It was corrected.

**(8) Line 211. Why 'correspondingly'?**

Yes, you are right, maybe the word 'correspondingly' is unnecessary here and can be omitted.

**(9) Line 236. Write dt with Delta as in the equation.**

'dt' was changed to' $\Delta t$'.

**(10). Line 253. Experiments. It is commendable to have run 10 ensemble members for each experiment but I cannot see that much use was made of the variability between them. It could be interesting to know how large this variability is. On this point, it is not clear if the SD in e.g. Figure 1 includes this or is just based on the zonal mean of the ensemble mean?**

Yes, we agree that the variability between ensemble members of the experiment is missing here. However, the std of iodine between ensemble members of the experiment is found to be less than 1%. We mentioned it in the text but due to it being extremely low it will not be seen in the figure, so we decided not to include it in the line's std. So, we plotted only the std of ensemble-mean iodine between tropical latitudes [$20^o$N - $20^o$S].

**(11) Line 261 'COMPARED to present-day'. Also, why is this a worst case? You cannot assume that. It is just an assumption to investigate the sensitivity.**

We use a 2-fold increase of iodine emissions as an assumption for a worst-case scenario compared to the present-day because the prognostic scenarios show less level of future iodine than 2 times of present-day level. At the same time, iodine abundance has tripled over the past 50 years (Cuevas et al. 2018 and Legrand et al. 2018). Nevertheless, we agree with your point that it is a rough estimate and in our study it is only used to assess sensitivity. We also could say that it

is the sensitivity with emissions worse than in the present time but it might not necessarily be "the worst of all".

Reference:

Cuevas, C., Maffezzoli, N., and Corella, J. e. a.: Rapid increase in atmospheric iodine levels in the North Atlantic since the mid-20th century., Nat Commun, 9, 1452, https://doi.org/10.1038/s41467-018-03756-1, 2018.

Legrand, M., McConnell, J. R., Preunkert, S., Arienzo, M., Chellman, N., Gleason, K., Sherwen, T., Evans, M. J., and Carpenter, L. J.: Alpine ice evidence of a three-fold increase in atmospheric iodine deposition since 1950 in Europe due to increasing oceanic emissions, Proceedings of the National Academy of Science, 115, 12 136–12 141, https://doi.org/10.1073/pnas.1809867115, 2018.

**(12) Line 264. 'we' -> 'were'?**

It was corrected.

**(13) Line 276. Figure 1 caption. Last line, why 'ozone'?**

Thank you, it was a mistake. Of course, there must be "iodine" there. It was corrected.

**(14) Lines 286 - 291. 'peculiarities' and text around this. This is not clear. The aim of GMD papers is to explain model behaviour and assumptions like this. The text needs clear rewriting to explain if the non-conservation is an issue and at what point it happens or is forced. The blue profile in Figure 1 looks very fixed on 1 pptv. Is that coincidental? The red line seems to show a bit more variation.**

Thank you for this comment and for pointing out this issue. We accept that the provided explanation here only reflects our assumptions regarding this issue but do not provide the solid and verified reasons for a gradual increase of iodine in the lower stratosphere seen in SOCOL. We performed several tests to reveal the reason for this. We firstly checked if there is a trend in source gases that might be a reason for the gradual increase seen in the lower stratosphere. Since organic emissions are the same for each year, we checked $HOI/I_2$ fluxes. The analysis did not reveal the trend. We can also speculate about relaxation time for iodine chemistry but we think that 10 years is enough to reach the equilibrium state. Nevertheless, we checked if there is still an "increase" of $I_y$ burden but comparing the level at 2000-2001 and 2008-2009 period, we found that it is not the case and the abundance of iodine is stable. Also, we tried to use an ideal-age traser to check if there is an issue in the model dynamics. The most possible reason for this increase is the removing of iodine species by interactive wet deposition (by convective cloud's rain) and/or effective/reactive uptake and removing/recycling on ice crystals that are still somehow affecting the transition zone in the lower stratosphere because, horizontally, the removing/recycling is not uniformly distributed since clouds are not everywhere presented and, therefore, stratospheric iodine loading is ubiquitously different. Thus, we could not find the exact process that is responsible for this gradual increase, and in the paper, we addressed only assumptions.

**(15) Line 300. The evaluation with the TORERO data is very crude. If there is a reason for this (e.g. free running CCM) then please state it. Why not sample the model like the observations? The 'doubling' assumes the same length of day/night. Roughly ok for the equator but this is just Jan/Feb so will be biased at other latitudes.**

Yes, we agree that the comparison against TORERO/CONTRAST observations is crudely performed. To make the evaluation fairer, we limited the comparison to only tropical latitudes [15°N-15°S] as the observations are rather scarce over other latitudes. Also, we sampled model data as observations to conduct an equitable comparison.

The new figure and analysis are addressed in the paper (the updated figure is presented below).

[Figure]

January/February averages of modeled and observed IO in the tropical troposphere [15°N-15°S] for (a) the TORERO campaign from Costa Rica (Jan./Feb. 2012, 10°N-40°S, 250°E-285°E), and (b) the CONTRAST campaign from Guam (Jan./Feb. 2014, 40°N-15°S, 115°E-175°E). Red line: IO from SOCOL-AERv2-I. Green line: IO from CAM-chem. Blue dots: IO observed by AMAX-DOAS. Shadings: IO standard deviations of all modeled/observed IO during the January-February period. Errorbars: AMAX-DOAS retrieval error.

**(16). As a general point the model evaluation with observations is very brief. What about data from other sources, e.g. the balloon and ground-based data mentioned in the introduction?**

Thank you for this comment. We agree that it would be desirable to add a comparison of iodine modeled by SOCOL with some of the local measurements too. So, we added the comparison of iodine compounds with some of the local measurements mentioned in the introduction as follows:

"The modeled reactive IO over Skandinavia (70°N; 20°E) in March is $> 0.45$ pptv at 17 km (monthly-mean value) that is in agreement with IO simulated with box model initialized partly with the IO retrieved by balloon flights (a day-time concentration is estimated to be $\sim 0.65$ pptv at 17 km) despite the measured upper limit of IO mixing ratio of 0.2 pptv (Pundt et al. 1998). SOCOL-AER2-I also captures well the $I_y$ estimated by box model (Pundt et al. 1998) showing a mixing ratio of about 1-1.1 pptv. Also, IO simulated with SOCOL-AERv2-I is corresponding well with DOAS measurements over Spitsbergen island (79°N; 12°E) in March (Wittrock et al. 2000) showing $> 0.48$ pptv in the lower stratosphere."